# Multi-objective Linear Reinforcement Learning with Lexicographic Rewards

**Bo Xue**[1]  **Dake Bu**[1]  **Ji Cheng**[1]  **Yuanyu Wan**[2]  **Qingfu Zhang**[1]

## Abstract

Reinforcement Learning (RL) with linear transition kernels and reward functions has recently attracted growing attention due to its computational efficiency and theoretical advancements. However, prior theoretical research in RL has primarily focused on single-objective problems, resulting in limited theoretical development for multi-objective reinforcement learning (MORL). To bridge this gap, we examine MORL under lexicographic reward structures, where rewards comprise $m$ hierarchically ordered objectives. In this framework, the agent maximizes objectives sequentially, prioritizing the highest-priority objective before considering subsequent ones. We introduce the first MORL algorithm with provable regret guarantees. For any objective $i \in \{1, 2, \ldots, m\}$, our algorithm achieves a regret bound of $\widetilde{O}(\Lambda^i(\lambda) \cdot \sqrt{d^2 H^4 K})$, where $\Lambda^i(\lambda) = 1 + \lambda + \cdots + \lambda^{i-1}$, $\lambda$ quantifies the trade-off between conflicting objectives, $d$ is the feature dimension, $H$ is the episode length, and $K$ is the number of episodes. Furthermore, our algorithm can be applied in the misspecified setting, where the regret bound for the $i$-th objective becomes $\widetilde{O}(\Lambda^i(\lambda) \cdot (\sqrt{d^2 H^4 K} + \epsilon d H^2 K))$, with $\epsilon$ denoting the degree of misspecification.

## 1. Introduction

Reinforcement Learning (RL) is a powerful paradigm wherein an agent learns to solve tasks through iterative trial and error with an unknown environment, dynamically adjusting its actions based on received reward signals (Sutton & Barto, 2018). This framework has proven effective in tackling complex decision-making challenges across various domains, including robotics (Kober et al., 2013), fi-

nance (Liu et al., 2022), healthcare (Yu et al., 2021), and autonomous systems (Merkle et al., 2024). Therefore, establishing a solid theoretical foundation for RL is crucial. Over the years, significant advancements have been made in understanding RL with linear transition kernels and reward functions, a subfield often referred to as linear RL or linear Markov Decision Process (MDP) (Zhang et al., 2024a; Zhou & Gu, 2022; Zhang et al., 2024b; Vial et al., 2022; He et al., 2021; Cassel et al., 2024; Xiong et al., 2023; Jin et al., 2020; Li & Sun, 2024; Li et al., 2024a). Nevertheless, these studies have primarily concentrated on single-objective problems, where a single reward function is optimized during the agent's interactions with the environment.

In contrast, many real-world decision-making problems require the simultaneous optimization of multiple, often conflicting objectives. For example, in autonomous driving, systems must simultaneously optimize safety and efficiency (Zhang et al., 2023), while energy management systems face trade-offs between cost and sustainability (Qiao et al., 2023). Traditional single-objective RL methods are are insufficient for handling the complexities arising from multiple objectives, motivating the development of Multi-objective Reinforcement Learning (MORL) (Gábor et al., 1998; Roijers et al., 2013; Abdolmaleki et al., 2020; Xu et al., 2020; Momma et al., 2022; Skalse et al., 2022; Cai et al., 2023).

Recent work in MORL extends conventional RL frameworks to accommodate multiple objectives through two primary approaches: (1) reward aggregation, which combine multiple objectives into a single objective using weighted sums or other scalarization techniques (Roijers et al., 2013). However, this method is highly sensitive to the choice of weights and *requires precise knowledge* of the relative importance of the objectives, which may not always be available or straightforward. (2) Pareto front construction, which assumes all objectives hold equal importance and employs Pareto dominance to identify Pareto-optimal decisions (Xu et al., 2020). Although this method does not require knowledge of objective importance, *it may lack practical applicability* in cases where the objectives are not of equal importance. For example, a hotel recommendation system must prioritize price, location, and service according to user preferences (Yager et al., 2011).

To overcome these limitations, a growing line of MORL

[1]Department of Computer Science, City University of Hong Kong, Hong Kong, China [2]School of Software Technology, Zhejiang University, Ningbo, China. Correspondence to: Qingfu Zhang <qingfu.zhang@cityu.edu.hk>.

*Proceedings of the 42nd International Conference on Machine Learning*, Vancouver, Canada. PMLR 267, 2025. Copyright 2025 by the author(s).

research has introduced lexicographic reward structures that enforce hierarchical optimization of objectives (Gábor et al., 1998; Wray & Zilberstein, 2015; Skalse et al., 2022; Tercan & Prabhu, 2024). This approach optimizes objectives hierarchically, ensuring that the most critical objectives are prioritized while still enabling effective optimization of secondary objectives. Such a hierarchical structure offers a more flexible and practical solution for decision-making in complex scenarios where the relative importance of objectives is partially known.

Although algorithmic advances has been made in MORL concerning scalarization methods (Roijers et al., 2013), Pareto front construction (Xu et al., 2020) and lexicographic rewards (Skalse et al., 2022), most of them heavily focused on empirical evaluations, with limited attention given to theoretical guarantees. This lack of analysis has hindered the development of principled algorithms with provable performance bounds. To bridge this gap, we leverage established theoretical foundations in single-objective linear reinforcement learning (He et al., 2023; Jin et al., 2020) and extend them to the finite-horizon MORL framework, specifically focusing on environments with linear transition kernels and reward functions. We term this framework as Multi-Objective Linear Reinforcement Learning (MOLRL). Our work provides a comprehensive theoretical analysis of regret bounds, thereby contributing to the development of more robust and scalable solutions in this field.

To the best of our knowledge, this is the first work to address MOLRL with lexicographic rewards. A key challenge in this setting is that the Bellman optimality equation (Sutton & Barto, 2018) does not hold, which invalidates greedy action-selection strategies, such as LSVI-UCB (Jin et al., 2020) and LSVI-UCB++ (He et al., 2023). We tackle this challenge and make the following contributions:

- We propose an MOLRL algorithm that decomposes the action-selection process into multiple stages, enabling a more refined balance between conflicting objectives. This approach prioritizes higher-level objectives while effectively managing lower-level ones.

- For the proposed algorithm, we derive a regret bound of $\widetilde{O}(\Lambda^i(\lambda) \cdot \sqrt{d^2 H^4 K})$ for any objective $i \in \{1, 2, \ldots, m\}$, where $m$ is the number of objectives and $\Lambda^i(\lambda) = 1 + \lambda + \cdots + \lambda^{i-1}$. $\lambda$ depicts the trade-off between conflicting objectives, $d$ is the feature vector dimension, $H$ denotes the episode length, and $K$ is the total number of episodes.

- Our algorithm can be applied in the misspecified MOLRL setting, where the transition kernel and reward function of the true environment are approximated by linear functions with a misspecification level of $\epsilon \geq 0$. In this setting, our algorithm achieves a regret bound

*Table 1.* Comparisons of the regret bounds for linear RL.

| Algorithm | Objectives | Regret |
|---|---|---|
| Jin et al. (2020) | Single | $\widetilde{O}(\sqrt{d^3 H^4 K})$ |
| Zanette et al. (2020) | Single | $\widetilde{O}(\sqrt{d^4 H^4 K} + d^4 H^5)$ |
| He et al. (2023) | Single | $\widetilde{O}(\sqrt{d^2 H^3 K} + d^7 H^8)$ |
| **This work** | Multiple | $\widetilde{O}(\Lambda^i(\lambda) \cdot \sqrt{d^2 H^4 K})$ |

of $\widetilde{O}(\Lambda^i(\lambda) \cdot (\sqrt{d^2 H^4 K} + \epsilon d H^2 K))$, maintaining robustness to model inaccuracies.

- As shown in Table 1, our algorithm achieves a regret bound comparable to single-objective RL algorithms in terms of the leading-order term $K$, while simultaneously optimizing multiple objectives.

## 2. Related Work

In this section, we briefly review the development of linear RL and multi-objective RL.

### 2.1. Linear Reinforcement Learning

The existing literature on linear reinforcement learning can be broadly categorized into two main research directions. The first direction focuses on establishing minimax-type regret bounds for proposed algorithms. Yang & Wang (2019) assumed linearity in both the reward function and transition probabilities and analyzed the complexity of finding an $\epsilon$-optimal policy. Later, Jin et al. (2020) focused on regret minimization, proposing a linear RL algorithm that achieves a regret bound of $\widetilde{O}(\sqrt{d^3 H^4 K})$. Their algorithm can also be applied to misspecified linear RL, attaining a regret bound of $\widetilde{O}(\sqrt{d^3 H^4 K} + \epsilon d H^2 K)$. In an improved approach, Vial et al. (2022) introduced a parameter-free algorithm for the misspecified linear RL problem, which does not require knowledge of the misspecification level $\epsilon$. Zanette et al. (2020) applied the Thompson sampling approach to linear RL, achieving a regret bound of $\widetilde{O}(\sqrt{d^4 H^5 K})$. He et al. (2023) developed a computationally efficient algorithm with a regret bound of $\widetilde{O}(\sqrt{d^2 H^3 K} + d^7 H^8)$ for linear RL.

The second research direction emphasizes deriving gap-dependent regret bounds for linear RL. He et al. (2021) established a regret bound of $\widetilde{O}(d^3 H^5 / \Delta \log(K))$ for the algorithm of Jin et al. (2020), where $\Delta$ is the minimal sub-optimality gap. More recently, Zhang et al. (2024a) proposed an improved algorithm that achieves a regret bound of $\widetilde{O}(d^3 H^5 / \Delta)$, which is independent of the episode number $K$. Additionally, other studies have explored linear RL in various settings. Zhou et al. (2021) achieved a nearly optimal minimax regret bound for RL with linear mixture MDPs, where the transition probability is a linear combination of several base models. Li et al. (2024b) analyzed dynamic

regret bounds for adversarial linear mixture MDPs, where optimal policies change over time. Dai et al. (2024) studied multi-agent linear RL, in which multiple agents share a common state space but have independent action spaces. However, these studies mainly address single-objective settings and do not extend to multi-objective problems.

## 2.2. Multi-objective Reinforcement Learning

Early approaches to MORL employed scalarization methods, which aggregate the reward vector into a single scalar through a weighted sum (Roijers et al., 2013; Van Moffaert et al., 2013; Agarwal et al., 2022). Although these methods are simple and intuitive, they require careful tuning of the weights (Abels et al., 2019; Yang et al., 2019). More recent MORL research has explored alternative strategies to address these challenges. For instance, Abdolmaleki et al. (2020) proposed encoding preferences via constraints on each objective, enabling the specification of desired preferences in a scale-invariant manner. Xu et al. (2020) incorporated the multi-objective policy gradient method into an evolutionary framework to identify a set of Pareto-optimal policies, where no objective can be improved without degrading another. Momma et al. (2022) introduced a general framework capable of incorporating multiple forms of preferences to identify Pareto-optimal policies.

In the context of MORL with lexicographic rewards, Gábor et al. (1998) first formalized this problem and introduced a greedy-based algorithm to solve it. Wray et al. (2015) introduced the slack-based Lexicographic RL, which allows for deviations from the optimal policy rather than adhering to absolute thresholds. More recently, Skalse et al. (2022) presented a family of algorithms based on action-value and policy gradient methods for solving lexicographic MORL. In the simplified setting of lexicographic bandits (i.e., RL with horizon $H = 1$), related work has been conducted on multi-armed bandits (Hüyük & Tekin, 2021), contextual bandits (Turgay et al., 2018), Lipschitz bandits (Xue et al., 2024), and linear bandits (Xue et al., 2025).

While many existing MORL studies focus primarily on empirical evaluations, several also offer valuable theoretical insights, such as convergence analyses and structural results (e.g., (Gábor et al., 1998)). However, formal regret guarantees, particularly in the setting of linear MDPs, remain underexplored. To address this gap, we present a comprehensive theoretical analysis of regret bounds within the MORL framework, thereby paving the way for more robust and scalable solutions in this domain.

## 3. Preliminaries

**Notation.** For any positive integer $N \in \mathbb{Z}_+$, we define $[N] \triangleq \{1, 2, \ldots, N\}$. The superscript $i \in [m]$ distinguishes

symbols associated with different objectives and the symbol $(\hat{\cdot})$ signifies that the variable is an estimate, e.g., $\hat{Q}^i(\cdot, \cdot)$ is the estimated action-value function for the $i$-th objective. The Euclidean norm of a vector $\phi \in \mathbb{R}^d$ is denoted by $\|\phi\|$. The norm induced by a positive-definite matrix $U \in \mathbb{R}^{d \times d}$ is defined by $\|\phi\|_U = \sqrt{\phi^\top U^{-1} \phi}$.

**MORL.** MORL is a framework in which an agent interacts with an environment to learn an optimal policy. Formally, MORL is modeled as a Multi-Objective Markov Decision Process (MOMDP), defined by the tuple $(\mathcal{S}, \mathcal{A}, H, \mathbb{P}, \boldsymbol{r})$, where $\mathcal{S}$ is the state space, $\mathcal{A}$ is the action space, $H \in \mathbb{Z}_+$ is the length of each episode. $\mathbb{P} = \{\mathbb{P}_h\}_{h=1}^H$ and $\boldsymbol{r} = \{\boldsymbol{r}_h\}_{h=1}^H$ are the transition probability functions and reward functions for each $h \in H$. Specifically, $\mathbb{P}_h(x'|x, a)$ represents the probability of transitioning from state $x \in \mathcal{S}$ to state $x' \in \mathcal{S}$ after taking action $a$ at step $h \in [H]$. The reward function at step $h$, $\boldsymbol{r}_h : \mathcal{S} \times \mathcal{A} \to [0, 1]^m$, is a deterministic function that maps state-action pair $(x, a)$ in to a reward vector

$$\boldsymbol{r}_h(x, a) = [r_h^1(x, a), r_h^2(x, a), \ldots, r_h^m(x, a)],$$

where $r_h^i(x, a)$ is the reward corresponding to the objective $i \in [m]$, and $m$ is the number of objectives.

A policy of an agent is a function $\pi : \mathcal{S} \times [H] \to \mathcal{A}$, where $\pi(x, h)$ specifies the action that the agent takes at state $x \in \mathcal{S}$ and step $h \in [H]$. For each objective $i \in [m]$ and state-action pair $(x, a) \in \mathcal{S} \times \mathcal{A}$, the action-value function $Q_{\pi, h}^i(x, a)$ (also known as the Q-function) and the value function $V_{\pi, h}^i(x)$ of a policy $\pi$ are defined as

$$Q_{\pi, h}^i(x, a) = r_h^i(x, a) +$$
$$\mathbb{E}\left[\sum_{h'=h+1}^H r_{h'}^i(x_{h'}, \pi(x_{h'}, h')) \mid x_h = x, a_h = a\right]$$
$$V_{\pi, h}^i(x) = \mathbb{E}\left[\sum_{h'=h}^H r_{h'}^i(x_{h'}, a_{h'}) \mid x_h = x\right]$$

Let $[\mathbb{P}_h V](x, a) = \mathbb{E}_{x' \sim \mathbb{P}_h(\cdot|x, a)}[V(x')]$ denote the expected value of function $V$ under the transition dynamics $\mathbb{P}_h$. The Bellman equation for objective $i \in [m]$ is

$$\begin{aligned} Q_{\pi, h}^i(x, a) &= r_h^i(x, a) + [\mathbb{P}_h V_{\pi, h+1}^i](x, a), \\ V_{\pi, h}^i(x) &= Q_{\pi, h}^i(x, \pi(x, h)). \end{aligned} \quad (1)$$

This formulation provides a recursive relationship between the Q-function and the value function, which is fundamental to the analysis and optimization of policies in RL.

**The Learning Process.** The learning process in MORL comprises $K$ episodes. At the start of each episode $k = 1, 2, \ldots, K$, the agent selects a policy $\pi_k$ based on data collected from the previous $k - 1$ episodes. The agent then interacts with the environment following $\pi_k$ over $H$ time

steps. At each step $h = 1, 2, \ldots, H$, the agent observes the environment state $x_{k,h}$ and selects an action $a_{k,h} = \pi_k(x_{k,h}, h)$. Subsequently, the environment transitions to a new state $x_{k,h+1} \sim \mathbb{P}_h(\cdot | x_{k,h}, a_{k,h})$ and provides the agent with a reward vector $\boldsymbol{r}_{k,h}$. After completing all $H$ steps, the agent collects the data $\{x_{k,h}, a_{k,h}, \boldsymbol{r}_{k,h}\}_{h=1}^H$ and uses it to refine its policy, thereby improving future decision-making.

How to evaluate policies in a multi-objective context remains an open issue. Since conflicting objectives cannot be maximized simultaneously, multi-objective work defines a notion of *dominance* to compare reward vectors and assess policies, such as Pareto dominance (Xu et al., 2020) and lexicographic dominance (Skalse et al., 2022). In this paper, we adopt the lexicographic order, where objectives are prioritized. The formal definition is as follows.

**Definition 1** (**Lexicographic Order**). *Consider two vectors* $\boldsymbol{u} = [u^1, \ldots, u^m], \boldsymbol{v} = [v^1, \ldots, v^m] \in \mathbb{R}^m$. $\boldsymbol{u}$ *lexicographically dominates* $\boldsymbol{v}$ *if and only if there exists some* $i^* \in [m]$ *such that* $u^i = v^i$ *for* $i \in [i^* - 1]$ *and* $u^{i^*} > v^{i^*}$.

Lexicographic order compares vectors element-wise, starting from the first and proceeding sequentially to the last. For example, $[5, 3, 4]$ lexicographically dominates $[5, 2, 5]$ and $[4, 8, 1]$. Based on this ordering, we define the lexicographically optimal policy (Skalse et al., 2022).

**Definition 2** (**Lexicographically Optimal Policy**). *A policy* $\pi_*$ *is lexicographically optimal if and only if, for any* $x \in \mathcal{S}$, *its vector of values* $[V_{\pi_*,1}^1(x), V_{\pi_*,1}^2(x), \ldots, V_{\pi_*,1}^m(x)]$ *is not lexicographically dominated by that of any other policy.*

A lexicographically optimal policy ensures that the most important objective is maximized, while still allowing some optimization of the lower-priority objectives.

Following single-objective RL (Jin et al., 2020), we evaluate the performance of the agent by regret, which quantifies the difference between the accumulated rewards of the agent's policy and those of a lexicographically optimal policy, i.e.,

$$R^i(K) = \sum_{k=1}^K V_{\pi_*,1}^i(x_{k,1}) - V_{\pi_k,1}^i(x_{k,1}), i \in [m].$$

To establish a regret bound for MORL, we quantify the trade-offs among conflicting objectives as follows.

**Assumption 1.** *Let* $\tilde{Q}_h^i(x, a) = r_h^i(x, a) + [\mathbb{P}_h \tilde{V}_{h+1}^i](x, a)$ *for any* $i \in [m]$ *and* $(x, a, h) \in S \times A \times [H]$. *Let* $\pi_*(x, h)$ *denote the action chosen by a lexicographically optimal policy at* $(x, h)$. *We assume* $\pi_*(x, h)$ *is the lexicographic optimal action for* $\tilde{Q}_h^i(x, a)$ *and the trade-off among objectives is governed by* $\lambda \geq 0$, *such that for all* $h \in [H]$ *and* $i \in [m]$,

$$\tilde{Q}_h^i(x, a) - \tilde{Q}_h^i(x, \pi_*(x, h))$$
$$\leq \lambda \cdot \max_{j \in [i-1]} \left\{ \tilde{Q}_h^j(x, \pi_*(x, h)) - \tilde{Q}_h^j(x, a) \right\}.$$

Here, $\tilde{V}_h^i(x) = \langle \boldsymbol{w}(x), \mathbf{r}_{h:H}^i \rangle$, *where* $\boldsymbol{w}(x) \in \mathbb{R}^{H-h+1}$ *is a shared weighting vector across all objectives, and* $\mathbf{r}_{h:H}^i = [r_h^i(\cdot, \cdot), r_{h+1}^i(\cdot, \cdot), \ldots, r_H^i(\cdot, \cdot)]$.

Here, $\lambda$ bounds the improvement in the value of the $i$-th objective for each unit decrease in the preceding $i - 1$ objectives. In multi-objective optimization, objectives often conflict, thus it is common to introduce parameters to quantify trade-offs, such as the global trade-off (Miettinen, 1999), the marginal rate of substitution (Miettinen, 1999), and the allowable trade-off (Wiecek, 2007).

In this paper, we focus on the linear MOMDP, a natural extension of linear MDP (Jin et al., 2020), defined as follows:

**Definition 3.** $\mathcal{M}(\mathcal{S}, \mathcal{A}, H, \mathbb{P}, \boldsymbol{r})$ *is a linear MOMDP with a feature map* $\phi : \mathcal{S} \times \mathcal{A} \to \mathbb{R}^d$, *if for any* $h \in [H]$, *there exist unknown measures* $\boldsymbol{\mu}_h(\cdot) : \mathcal{S} \to \mathbb{R}^d$ *and unknown vectors* $\boldsymbol{\theta}_h^i \in \mathbb{R}^d$, *such that for any* $(x, a) \in \mathcal{S} \times \mathcal{A}$,

$$\mathbb{P}_h(\cdot | x, a) = \langle \phi(x, a), \boldsymbol{\mu}_h(\cdot) \rangle, r_h^i(x, a) = \langle \phi(x, a), \boldsymbol{\theta}_h^i \rangle,$$

*where* $\|\phi(x, a)\| \leq 1$ *for all* $(x, a) \in \mathcal{S} \times \mathcal{A}$, *and* $\max\{\|\boldsymbol{\mu}_h(\cdot)\|, \|\boldsymbol{\theta}_h^i\|\} \leq \sqrt{d}$ *for all* $h \in [H]$ *and* $i \in [m]$.

In environments with large state and action spaces, linear structure significantly reduces computational complexity, and well-established mathematical techniques and tools for the liner model facilitate rigorous analysis. Regarding the relationship between general and linear MDP, any MDP with a finite state space $\mathcal{S}$ and action space $\mathcal{A}$ can be represented as a linear MDP by encoding each state-action pair $(x, a)$ as a one-hot feature vector in $\mathbb{R}^d$, where $d = |\mathcal{S}| \times |\mathcal{A}|$. The transition kernel $\mathbb{P}_h(\cdot | x, a)$ and reward function $r_h^i(x, a)$ can be expressed as inner products between the feature vector of $(x, a)$ and learnable parameters.

## 4. Algorithm

In this section, we present the Lexicographic Linear Reinforcement Learning (LLRL) algorithm, which effectively combines the computational efficiency of linear MOMDPs with the hierarchical prioritization inherent in lexicographic rewards, ensuring that higher-priority objectives are maximized before considering lower-priority ones.

**Initialization.** LLRL learns the optimal policy over $K$ episodes, with each episode comprises $H$ interaction steps with the environment. Initially, the covariance matrix $\mathbf{U}_h$ is initialized as the identity matrix $\mathbf{I} \in \mathbb{R}^{d \times d}$ for all $h \in [H]$. Similarly, the Q-functions $\hat{Q}_h^i(\cdot, \cdot)$ are initialized to zero for all objectives $i \in [m]$ and time steps $h \in [H]$. Both the covariance matrix and Q-functions are updated iteratively across episodes based on the rewards received at each step. At the terminal step $H + 1$, the Q-functions for all objectives are set to zero, as no further rewards can be obtained.

**Algorithm 1** Lexicographic Linear Reinforcement Learning

**Input:** $\delta, d, m, K, H, \lambda, \{\beta_k\}_{k\in[K]}$

1: Initialize $U_h = I, \hat{Q}_h^1(\cdot,\cdot) = \cdots = \hat{Q}_h^m(\cdot,\cdot) = 0, \forall h \in [H], \hat{Q}_{H+1}^1(\cdot,\cdot) = \cdots = \hat{Q}_{H+1}^m(\cdot,\cdot) = 0$
2: **for** $k = 1, 2, \ldots, K$ **do**
3:    Receive the initial state $x_{k,1}$
4:    **for** $h = 1, 2, \ldots, H$ **do**
5:       Initialize $s = 1, \mathcal{A}_s = \mathcal{A}$
6:       **repeat**
7:          **if** $\|\phi(x_{k,h}, a)\|_{U_h} \leq 1/\sqrt{K}, \forall a \in \mathcal{A}_s$ **then**
8:            Run Algorithm 2 to refine actions: $\mathcal{A}_K = \text{LAE}(\{\hat{Q}_h^i(x_{k,h}, \cdot)\}_{i\in[m]}, \beta_k, \mathcal{A}_s, 1/\sqrt{K})$
9:            Randomly take an action $a_{k,h} \in \mathcal{A}_K$
10:         **else if** $\|\phi(x_{k,h}, a_{k,h})\|_{U_h} > 2^{-s}, \exists a_{k,h} \in \mathcal{A}_s$ **then**
11:            Take the action $a_{k,h}$
12:         **else**
13:            Run Algorithm 2 to refine actions: $\mathcal{A}_{s+1} = \text{LAE}(\{\hat{Q}_h^i(x_{k,h}, \cdot)\}_{i\in[m]}, \beta_k, \mathcal{A}_s, 2^{-s})$
14:            Update $s = s + 1$
15:         **end if**
16:       **until** an action $a_{k,h}$ is taken
17:       Observe rewards $\boldsymbol{r}_{k,h}$ and next state $x_{k,h+1}$
18:    **end for**
19:    **for** $h = H, H-1, \ldots, 1$ **do**
20:       $U_h = \sum_{\tau=1}^{k} \phi(x_{\tau,h}, a_{\tau,h})\phi(x_{\tau,h}, a_{\tau,h})^\top + I$
21:       $\hat{r}_{\tau,h}^i = r_{\tau,h}^i + \hat{Q}_{h+1}^i(x_{\tau,h+1}, a_{\tau,h+1}), \forall \tau \in [k]$ and $i \in [m]$
22:       $\hat{\boldsymbol{w}}_h^i = U_h^{-1} \sum_{\tau=1}^{k} \phi(x_{\tau,h}, a_{\tau,h}) \cdot \hat{r}_{\tau,h}^i, \forall i \in [m]$
23:       Update the estimated Q-function: $\hat{Q}_h^i(x, a) = \langle \hat{\boldsymbol{w}}_h^i, \phi(x, a) \rangle, \forall (x, a) \in \mathcal{S} \times \mathcal{A}$ and $i \in [m]$
24:    **end for**
25: **end for**

---

**Decision-making.** In each episode, LLRL begins by receiving an initial state $x_{k,1}$ and then enters into $H$ interaction steps. At each step $h = 1, 2, \ldots, H$, LLRL selects an action by balancing exploration (i.e., acquiring information about rarely observed actions) and exploitation (i.e., choosing the most promising action). To manage trade-offs among different objectives, the decision-making process is *divided into multiple stages*. Actions are refined based on uncertainty checks and lexicographic prioritization.

Before the decision-making process begins, LLRL initializes the stage index $s = 1$ and the candidate action set $\mathcal{A}_s = \mathcal{A}$. Then, LLRL iteratively refines the candidate actions until a final action is chosen. For the given state $x_{k,h}$ of current step, LLRL considers three scenarios:

(i) If $\|\phi(x_{k,h}, a)\|_{U_h} \leq \frac{1}{\sqrt{K}}$ for all $a \in \mathcal{A}_s$, this indicates that *all actions in $\mathcal{A}_s$ have been sufficiently explored*. In this case, LLRL invokes the Lexicographic Action Elimination

**Algorithm 2** Lexicographic Action Elimination

**Input:** $\{\hat{Q}_h^i(x_{k,h}, \cdot)\}_{i\in[m]}, \beta_k, \mathcal{A}_s, C$

1: Initialize $\mathcal{A}_s^0 = \mathcal{A}_s$
2: **for** $i = 1, 2, \ldots, m$ **do**
3:    $\hat{a}_{k,h}^i = \text{argmax}_{a \in \mathcal{A}_s^{i-1}} \hat{Q}_h^i(x_{k,h}, a)$
4:    $\mathcal{A}_s^i = \{a \in \mathcal{A}_s^{i-1} | \hat{Q}_h^i(x_{k,h}, \hat{a}_{k,h}^i) - \hat{Q}_h^i(x_{k,h}, a) \leq (2 + 4\lambda + \cdots + 4\lambda^{i-1}) \cdot \beta_k \cdot C\}$
5: **end for**
6: Return $\mathcal{A}_s^m$

---

(LAE) algorithm (Algorithm 2) to refine the candidate action set, producing a smaller subset of promising actions $\mathcal{A}_K$. An action $a_{k,h}$ is then randomly selected from $\mathcal{A}_K$. LAE uses the estimated Q-functions $\hat{Q}_h^i(x_{k,h}, \cdot)$ and the uncertainty bound $1/\sqrt{K}$ to sequentially eliminate suboptimal actions from $\mathcal{A}_s$ by the priority of objectives. Further details on the LAE are provided in the following content.

(ii) If $\|\phi(x_{k,h}, a_{k,h})\|_{U_h} > 2^{-s}$ for some action $a_{k,h} \in \mathcal{A}_s$, this action is directly selected due to *its high uncertainty*, which requires further exploration.

(iii) If all actions $a \in \mathcal{A}_s$ satisfy $\|\phi(x_{k,h}, a)\|_{U_h} \leq 2^{-s}$, it implies that *the uncertainty in the estimated Q-functions for all $a \in \mathcal{A}_s$ is bounded by $2^{-s}$*. In this case, LAE is invoked again to refine the candidate set, using the uncertainty bound $2^{-s}$ as the exploration threshold. The stage index $s$ is then incremented to $s + 1$ for a more refined elimination process.

This iterative process continues until an action is selected. Once an action $a_{k,h}$ is chosen, LLRL observes the corresponding reward $\boldsymbol{r}_{k,h} = [r_{k,h}^1, r_{k,h}^2, \ldots, r_{k,h}^m]$ and transitions to the next state $x_{k,h+1}$. The episode concludes after all $H$ steps are completed. The intuition behind this policy design is further discussed in Section 6.

**Lexicographic Action Elimination.** LAE plays a crucial role in balancing trade-offs among conflicting objectives. It iteratively refines the action set by prioritizing objectives sequentially, from the first to the last. Initially, LAE defines the action set as $\mathcal{A}_s^0 = \mathcal{A}_s$, which contains all candidate actions of the current stage. For each objective $i \in [m]$, LAE selects the action $\hat{a}_{k,h}^i \in \mathcal{A}_s^{i-1}$ that maximizes the estimated Q-value $\hat{Q}_h^i(x_{k,h}, a)$, i.e.,

$$\hat{a}_{k,h}^i = \underset{a \in \mathcal{A}_s^{i-1}}{\text{argmax}} \hat{Q}_h^i(x_{k,h}, a).$$

This action, $\hat{a}_{k,h}^i$, serves as the reference point to eliminate suboptimal actions. Precisely, LAE removes actions whose estimated Q-values deviate significantly from $\hat{a}_{k,h}^i$, i.e.,

$$\mathcal{A}_s^i = \{a \in \mathcal{A}_s^{i-1} | \hat{Q}_h^i(x_{k,h}, \hat{a}_{k,h}^i) - \hat{Q}_h^i(x_{k,h}, a) \leq (2 + 4\lambda + \cdots + 4\lambda^{i-1}) \cdot \beta_k \cdot C\},$$

where $C$ is a dynamically adjusted threshold to balance exploration and exploitation. Specifically, $C = 1/\sqrt{K}$ in case (i), and $C = 2^{-s}$ in case (iii) of the $s$-th stage.

After processing all $m$ objectives, the final set $\mathcal{A}_s^m$ consists of actions deemed promising across all objectives and can be returned to LLRL for a more refined decision-making process in the next stage $s + 1$.

**Policy Update.** In each episode $k$, after completing its interactions with the environment, the agent gathers the dataset $\{x_{k,h}, a_{k,h}, \boldsymbol{r}_{k,h}\}_{h=1}^H$ and initiates a backward update procedure. Owing to the linear structure of the linear MDP, the inherent Q-function $Q_{\pi,h}^i(\cdot, \cdot)$ for any objective $i \in [m]$ and policy $\pi$ exhibits linearity with respect to the feature vector $\phi(x, a)$. This property is formally expressed in the following proposition, which generalizes Proposition 2.3 of Jin et al. (2020) to a multi-objective context.

**Proposition 1.** *In a linear MOMDP, for any objective $i \in [m]$ and any policy $\pi$, there exist weights $\{\boldsymbol{w}_{\pi,h}^i\}_{h \in [H]}$ such that for any $(x, a, h) \in \mathcal{S} \times \mathcal{A} \times [H]$, we have*

$$Q_{\pi,h}^i(x, a) = \langle \phi(x, a), \boldsymbol{w}_{\pi,h}^i \rangle.$$

Based on Proposition 1, LLRL estimates $Q_{\pi,h}^i(x, a)$ following the procedure of least-squares estimation. According to the Bellman equation (Eq. (1)), the Q-function at the $h$-th step $Q_{\pi,h}^i$ is influenced by the future value function $V_{\pi,h+1}^i$. Therefore, the estimation process starts from the final step $h = H$ and proceeds backward to the first step $h = 1$.

During the backward pass, for each step $h = H, \ldots, 1$, LLRL first updates the covariance matrix $\mathrm{U}_h$ using the feature vectors observed at the $h$-th step of previous episodes:

$$\mathrm{U}_h = \sum_{\tau=1}^k \phi(x_{\tau,h}, a_{\tau,h}) \phi(x_{\tau,h}, a_{\tau,h})^\top + \mathrm{I}. \quad (2)$$

Subsequently, to perform least-squares estimation for objective $i \in [m]$, LLRL combines the immediate reward $r_{k,h}^i$ and the future Q-value $\hat{Q}_{h+1}^i(x_{k,h+1}, a_{k,h+1})$ to form a surrogate reward, given by:

$$\hat{r}_{k,h}^i = r_{k,h}^i + \hat{Q}_{h+1}^i(x_{k,h+1}, a_{k,h+1}). \quad (3)$$

This bootstrapping procedure integrates immediate rewards with long-term Q-values, aligning with the principle of Bellman equation (1).

LLRL then estimates the parameter $\boldsymbol{w}_{\pi,h}^i$ for objective $i \in [m]$ using least-squares estimation, formulated as:

$$\hat{\boldsymbol{w}}_h^i = \underset{\boldsymbol{w} \in \mathbb{R}^d}{\arg\min} \sum_{\tau=1}^k \left( \langle \phi(x_{\tau,h}, a_{\tau,h}), \boldsymbol{w} \rangle - \hat{r}_{\tau,h}^i \right)^2$$
$$+ \|\boldsymbol{w}\|^2 \quad (4)$$
$$= \mathrm{U}_h^{-1} \sum_{\tau=1}^k \hat{r}_{\tau,h}^i \cdot \phi(x_{\tau,h}, a_{\tau,h}).$$

Finally, the estimated Q-function $\hat{Q}_h^i(\cdot, \cdot)$ for objective $i \in [m]$ is updated using the newly obtained parameter $\hat{\boldsymbol{w}}_h^i$, i.e.,

$$\hat{Q}_h^i(x, a) = \langle \hat{\boldsymbol{w}}_h^i, \phi(x, a) \rangle, \forall (x, a) \in \mathcal{S} \times \mathcal{A}. \quad (5)$$

The steps outlined in Eqs. (2), (3), (4), and (5) constitute the core of the backward pass in the LLRL algorithm. By iteratively updating the Q-functions of each objective, LLRL reduces uncertainty in parameter estimation and progressively improves the agent's policies over time.

## 5. Theoretical Guarantees

In this section, we provide the regret bounds for LLRL to rigorously analyze its performance. We establish two theorems that offer theoretical guarantees for different models. Specifically, Theorem 1 characterizes the performance of LLRL in the linear MOMDP setting, while Theorem 2 examines its behavior in the $\epsilon$-approximate linear MOMDP. These results not only validate the reliability of LLRL across different scenarios but also provide insights into its limitations.

**Theorem 1.** *Let $\Lambda^i(\lambda) = 1 + \lambda + \cdots + \lambda^{i-1}$ for $i \in [m]$. For a linear MOMDP, if LLRL is run with $\beta_k = 3H\sqrt{d \log\left(\frac{2mkH}{\delta}\right)}$, then with probability at least $1 - 2\delta$, the regret for any objective $i \in [m]$ satisfies*

$$R^i(K) \leq \widetilde{O}\left( \Lambda^i(\lambda) \cdot \sqrt{d^2 H^4 K} \right).$$

**Remark 1.** Theorem 1 states that for linear MOMDPs with lexicographic rewards, LLRL achieves a regret bound of $\widetilde{O}\left( \Lambda^i(\lambda) \cdot \sqrt{d^2 H^4 K} \right)$ for any objective $i \in [m]$. This matches the near-optimal dependence on feature dimension $d$ and time horizon $K$ observed in single-objective RL (He et al., 2023). The multiplicative factor $\Lambda^i(\lambda)$ quantifies the trade-off in lexicographic optimization: for the primary objective, $\Lambda^1(\lambda) = 1$ leads to an improved bound compared to single-objective baselines (Jin et al., 2020) in terms of $d$, while lower-priority objectives incur increasing regret as $\Lambda^i(\lambda)$ grows with their position $i$ in the priority hierarchy.

Theorem 1 relies on the linear structure of the MDP. However, real-world applications often involve nonlinear MDPs, leading to potential misspecification when assuming linearity. To address this, we first introduce an approximate linear model and then establish the regret bounds of LLRL under such misspecified conditions.

**Definition 4.** $\mathcal{M}(\mathcal{S}, \mathcal{A}, H, \mathbb{P}, \boldsymbol{r})$ *is an $\epsilon$-approximate linear MOMDP with a feature map $\phi : \mathcal{S} \times \mathcal{A} \to \mathbb{R}^d$, if for any $h \in [H]$, there exist unknown measures $\boldsymbol{\mu}_h(\cdot) : \mathcal{S} \to \mathbb{R}^d$ and unknown vectors $\boldsymbol{\theta}_h^i \in \mathbb{R}^d$, such that for any $(x, a) \in \mathcal{S} \times \mathcal{A}$,*

$$\|\mathbb{P}_h(\cdot | x, a) - \langle \phi(x, a), \boldsymbol{\mu}_h(\cdot) \rangle\|_1 \leq \epsilon,$$
$$|r_h^i(x, a) - \langle \phi(x, a), \boldsymbol{\theta}_h^i \rangle| \leq \epsilon, \quad (6)$$

where $\|\phi(x, a)\| \leq 1$ *for all* $(x, a) \in \mathcal{S} \times \mathcal{A}$, *and* $\max\{\|\boldsymbol{\mu}_h(\cdot)\|, \|\boldsymbol{\theta}_h^i\|\} \leq \sqrt{d}$ *for all* $h \in [H]$ *and* $i \in [m]$.

**Theorem 2.** *Let* $\Lambda^i(\lambda) = 1 + \lambda + \cdots + \lambda^{i-1}$ *for* $i \in [m]$. *For an* $\epsilon$-*approximate linear MOMDP, if LLRL is run with* $\beta_k = 3H\sqrt{d \log\left(\frac{2mkH}{\delta}\right)} + 2H\epsilon\sqrt{kd}$ *and the threshold* $C$ *in LAE is modified by* $C + 2H\epsilon$, *then with probability at least* $1 - \delta$, *the regret for any objective* $i \in [m]$ *satisfies*

$$R^i(K) \leq \widetilde{O}\left(\Lambda^i(\lambda) \cdot \left(\sqrt{d^2 H^4 K} + \epsilon d H^2 K\right)\right).$$

**Remark 2.** Theorem 2 states that for $\epsilon$-approximate linear MOMDPs with lexicographic rewards, LLRL achieves a regret bound of $\widetilde{O}\left(\Lambda^i(\lambda) \cdot \left(\sqrt{d^2 H^4 K} + \epsilon d H^2 K\right)\right)$ for any objective $i \in [m]$. This matches the dependence on $\epsilon$, $H$, and $K$ observed in single-objective RL (Jin et al., 2020), while improving the dependence on $d$. The linear model serves as an approximation of the true environment. Compared to Theorem 1, the additional regret term $\widetilde{O}(\epsilon d H^2 K)$ stems from approximation error, which decreases as model accuracy improves.

**Complexity Analysis.** We provide a detailed complexity analysis of Algorithm 1 and discuss its computational limitations. In Step 7, the complexity is $O(d^2|\mathcal{A}|)$, while Step 8 requires $O(md|\mathcal{A}|)$ computations due to the LAE procedure. Step 22 incurs a computational cost of $O(mkd + md^2)$, primarily driven by the matrix inversion of $U_h$ ($O(d^2)$) and the computation of $m$ linear regressions (i.e., $O(mkd + md^2)$). Summing over all $H$ layers of the MDP, the overall complexity becomes $O(Hmd|\mathcal{A}| + Hd^2|\mathcal{A}| + Hmkd + Hmd^2)$. When aggregated over $K$ rounds, the total computational complexity is $O(KHd|\mathcal{A}|(m+d) + K^2Hmd + KHmd^2)$. Notably, the $O(K^2)$ term results in significantly higher complexity compared to standard bandit algorithms (Xue et al., 2025). Nevertheless, our approach remains competitive with existing methods for single-objective MDP (Jin et al., 2020; Zanette et al., 2020; He et al., 2023).

**Further Improvements.** As summarized in Table 1, our algorithm's regret bound remains slightly higher than the near-optimal $\widetilde{O}(\sqrt{d^2 H^3 K} + d^7 H^8)$ bound for single-objective linear MDPs in terms of $H$ (He et al., 2023). This gap does not stem from a fundamental limitation but our current policy update mechanism, which employs a simpler approach compared to advanced methods. By incorporating the LSVI-UCB++ update framework (He et al., 2023), we anticipate refining LLRL to achieve an improved regret bound of $\widetilde{O}\left(\Lambda^i(\lambda) \cdot \left(\sqrt{d^2 H^3 K} + d^7 H^8\right)\right)$. We plan to investigate this promising direction in future research.

# 6. Challenges and Key Techniques

In this section, we analyze the main challenges in designing algorithms for linear MOMDPs with lexicographic rewards

and highlight our techniques to resolve them.

## 6.1. Challenges

**Optimal Action Preservation Dilemma.** The first challenge in the lexicographic setting is how to keep the optimal action during action elimination procedures. Lexicographic RL requires sequential elimination across objectives, but uncertainty in transition dynamics $\mathbb{P}_h(\cdot|x, a)$ brings optimal action preservation risks during confidence interval-based elimination. Precisely, when addressing the $i$-th objective, the agent constructs the confidence intervals of Q-values and eliminates actions whose Q-value intervals do not overlap with that of the most promising action. In the lexicographic setting, however, the confidence interval of the lexicographically optimal action may not overlap with that of the most promising action, leading to the loss of the optimal action. An example is provided to illustrate this issue.

**Example 1.** *Suppose there are three Q-value vectors:* $[5, 5, 5], [1, 5, 5]$ *and* $[4, 10, 1]$ *for actions* $a_1, a_2$ *and* $a_3$, *respectively.* $a_1$ *is the lexicographically optimal action. When eliminating actions based on the first objective,* $a_2$ *is eliminated because* $1$ *is far from* $5$, *but* $a_3$ *is kept as* $4$ *is close to* $5$. *After this step, only actions* $\{a_1, a_3\}$ *are contained in the candidate action. Next, the agent proceeds to eliminate actions based on the second objective. Since* $10$ *is much bigger than* $5$, *thus the optimal action* $a_1$ *is eliminated, which is disappointing because* $a_3$ *is awful for the third objective.*

This demonstrates that standard elimination techniques may discard the lexicographically optimal action, necessitating novel preservation mechanisms.

**Failure of Bellman Optimality.** Another challenge is to develop a decision-making approach that progressively converges to the optimal policy. In single-objective RL, the Bellman optimality equation serves as a foundation for algorithm design and theoretical analysis. It characterizes the optimal value function through an intuitive decision-making process and a recursive relationship, i.e.,

$$\begin{aligned} V_{\pi_*, h}(x) &= \max_{a \in \mathcal{A}} Q_{\pi_*, h}(x, a), \\ Q_{\pi_*, h}(x, a) &= r_h(x, a) + [\mathbb{P}_h V_{\pi_*, h+1}](x, a), \end{aligned} \quad (7)$$

where we omit the superscript for single-objective notation. This framework enables straightforward policies like:

$$a_{k,h} = \underset{a \in \mathcal{A}}{\operatorname{argmax}} \hat{Q}_h(x_{k,h}, a), \quad (8)$$

as employed in LSVI-UCB (Jin et al., 2020) and LSVI-UCB++ (He et al., 2023).

In the lexicographic setting, this structure breaks down for secondary objectives, as shown in the following example.

**Example 2.** *Let* $[Q_{\pi_*, h}^1(x, a_1), Q_{\pi_*, h}^2(x, a_1)] = [2, 3]$ *and* $[Q_{\pi_*, h}^1(x, a_2), Q_{\pi_*, h}^2(x, a_2)] = [1, 4]$ *for actions* $a_1, a_2 \in$

$\mathcal{A}$ and $x \in \mathcal{S}$. Since $[2, 3]$ *lexicographically dominates* $[1, 4]$, *the lexicographically optimal action* $\pi_*(x, h) = a_1$. *According to the Bellman equation* (1), *we obtain* $V^2_{\pi_*, h}(x) = Q^2_{\pi_*, h}(x, \pi_*(x, h)) = 3$. *However, for the second objective, the maximum Q-value is* $\max_{a \in \{a_1, a_2\}} Q^2_{\pi_*, h}(x, a) = 4$, *which does not equal to* $V^2_{\pi_*, h}(x)$.

This example establishes the following proposition.

**Proposition 2.** *There exists an MOMDP with lexicographic rewards problem, where for some* $(x, a) \in \mathcal{S} \times \mathcal{A}$ *and* $i \in [m]$,
$$V^i_{\pi_*, h}(x) \neq \max_{a \in \mathcal{A}} Q^i_{\pi_*, h}(x, a)$$
*for the lexicographically optimal poliy* $\pi_*$.

This divergence from Bellman optimality complicates both algorithm design and theoretical analysis, precluding the direct methods like Eq. (8).

### 6.2. Key Techniques

In this part, we highlight the key techniques used to address the aforementioned challenges.

For clarity, we denote $U_{k,h}$, $\hat{\boldsymbol{w}}^i_{k,h}$, and $\hat{Q}^i_{k,h}$ as the parameters and the estimated Q-function at episode $k$, with $\pi_k$ representing the policy at $k$-th episode. The estimated value function $\hat{V}^i_{k,h}$ is defined as $\hat{V}^i_{k,h}(x) = \hat{Q}^i_{k,h}(x, \pi_k(x, h))$. For any $(x, a, k, h) \in \mathcal{S} \times \mathcal{A} \times [K] \times [H]$, we define the surrogate Q-function for objective $i \in [m]$ as

$$\bar{Q}^i_{k,h}(x, a) = r^i_h(x, a) + \int_{x' \in \mathcal{S}} \hat{V}^i_{k,h+1}(x') \mathbb{P}_h(x'|x, a) \mathrm{d}x'.$$

Since $r^i_h(x, a)$ and $\mathbb{P}_h(x'|x, a)$ depend on the unknown parameters $\boldsymbol{\theta}^i_h$ and $\boldsymbol{\mu}_h(x')$, respectively, we first provide confidence intervals for the surrogate Q-functions.

**Lemma 1.** *Let* $\beta_k = 3H\sqrt{d \log\left(\frac{2mkH}{\delta}\right)}$. *With probability at least* $1 - \delta$, *for all* $(x, a, k, h) \in \mathcal{S} \times \mathcal{A} \times [K] \times [H]$ *and* $i \in [m]$, *we have*
$$|\langle \phi(x, a), \hat{\boldsymbol{w}}^i_{k,h} \rangle - \bar{Q}^i_{k,h}(x, a)| \leq \beta_k \|\phi(x, a)\|_{U_{k,h}}.$$

Building on this, we analyze the discrepancy between estimated and inherent value functions under policy $\pi_k$:

**Lemma 2.** *Let* $\delta^i_{k,h} = \hat{V}^i_{k,h}(x_{k,h}) - V^i_{\pi_k, h}(x_{k,h})$ *and* $\zeta^i_{k,h} = \delta^i_{k,h} - \mathbb{E}[\delta^i_{k,h}|x_{k,h}, a_{k,h}]$. *With probability at least* $1 - \delta$, *for any* $(k, h) \in [K] \times [H]$ *and* $i \in [m]$, *we have*

$$\hat{V}^i_{k,h}(x_{k,h}) - V^i_{\pi_k, h}(x_{k,h})$$
$$\leq \sum_{h'=h}^{H} 2\beta_k \cdot 2^{-s_{k,h'}(x_{k,h'})} + \sum_{h'=h+1}^{H} \zeta^i_{k,h'},$$

*where* $s_{k,h'}(x_{k,h'})$ *denotes the stage at which LLRL selects the action for state* $x_{k,h'}$.

Now, we highlight how to avoid losing the lexicographically optimal action. Lemma 1 reveals that all objectives share a common confidence term $\beta_k \|\phi(x, a)\|_{U_{k,h}}$, a distinctive feature of MOMDPs. To ensure the overlap between confidence intervals of lexicographically optimal actions and the most promising action of current round, we scale the confidence term. As detailed in Algorithm 2, the confidence term $C$ is scaled by the following factor:

$$2 + 4\lambda + \cdots + 4\lambda^{i-1},$$

which is specially designed to guarantee the confidence intervals of the lexicographic optimal action and the most promising action of current round overlap. Therefore, the lexicographically optimal action is preserved, as formalized in the following lemma:

**Lemma 3.** *For Algorithm 2, if* $\pi_*(x, h) \in \mathcal{A}_s$, *then with probability at least* $1 - \delta$, *for any objective* $i \in [m]$ *and action* $a \in \mathcal{A}^i_s$, *we have*

$$\pi_*(x, h) \in \mathcal{A}^i_s, \text{ and}$$
$$\bar{Q}^i_{k,h}(x, \pi_*(x, h)) - \bar{Q}^i_{k,h}(x, a) \leq 4\Lambda^i(\lambda) \cdot \beta_k \cdot C.$$

Finally, we highlight the basic idea of dealing with the failure of Bellman optimality in MORL. Since the Bellman optimality equation does not hold, direct maximization of Q-values fails to recover lexicographically optimal actions due to conflicting objectives and environment uncertainty. As an alternative, LLRL initiates a multi-stage action refinement process, which progressively eliminates suboptimal actions using geometrically decreasing confidence thresholds $2^{-s}$. This mechanism operates in three regimes: 1) Aggressive pruning under low uncertainty, 2) Exploration prioritization for high-uncertainty actions $\|\phi(x, a)\|_{U_h} > 2^{-s}$, and 3) Confidence-constrained elimination via the LAE subroutine for intermediate uncertainty $\mathcal{A}_{s+1} = \mathrm{LAE}(\cdot, 2^{-s})$. Crucially, lexicographic optimality is preserved through overlapping confidence intervals, ensuring the retention of lexicographically optimal actions despite Bellman equation violations. By replacing Q-maximization with a *threshold-adaptive elimination paradigm*, LLRL decouples action selection from Bellman optimality assumptions while controlling cumulative suboptimality via geometrically tightening thresholds. The resultant value function gap, formalized in Lemma 4, quantifies how uncertainty reduction across stages drives convergence to near-optimal policies.

**Lemma 4.** *With probability at least* $1 - \delta$, *for all* $(x, a, k, h) \in \mathcal{S} \times \mathcal{A} \times [K] \times [H]$ *and* $i \in [m]$, *we have*

$$V^i_{\pi_*, h}(x_{k,h}) - \hat{V}^i_{k,h}(x_{k,h}) \leq \sum_{h'=h}^{H} 10\Lambda^i(\lambda) \cdot \beta_k \cdot 2^{-s_{k,h'}(x_{k,h'})}$$

*where* $s_{k,h'}(x_{k,h'})$ *denotes the stage at which LLRL selects the action for state* $x_{k,h'}$.

As the learning process advances, the uncertainty in value estimation diminishes geometrically, evidenced by the exponential decay of the term $2^{-s_{k,h'}(x_{k,h'})}$. This ensures the estimated value functions $\hat{V}_{k,h}^i$ converge toward their optimal counterparts $V_{\pi_*,h}^i$. By combining Lemma 2 and Lemma 4, we can bound the per-episode instantaneous regret $V_{\pi_*,h}^i(x_{k,1}) - V_{\pi_k,h}^i(x_{k,1})$ at each episode $k \in [K]$, leading to the regret bound $\widetilde{O}(\Lambda^i(\lambda) \cdot \sqrt{d^2 H^4 K})$ in Theorem 1. Please refer to appendix for more details.

## 7. Conclusion and Future Work

This work establishes the first theoretical regret guarantee for MORL. We propose the LLRL algorithm, which addresses the failure of Bellman optimality through a *multistage confidence-guided elimination* strategy. By decomposing action selection into refined phases with geometrically decaying thresholds $2^{-s}$, LLRL attains a regret bound of $\widetilde{O}(\Lambda^i(\lambda) \cdot \sqrt{d^2 H^4 K})$ for all objectives $i \in [m]$, matching single-objective baselines (Jin et al., 2020; He et al., 2023) in leading $K$-dependence. Meanwhile, $\Lambda^1(\lambda) = 1$ indicates LLRL recover the near-optimal regret bounds of He et al. (2023) for the first objective in terms of both $d$ and $K$. Furthermore, LLRL naturally extends to misspecified settings through minor algorithmic modifications, maintaining $\widetilde{O}(\Lambda^i(\lambda) \cdot (\sqrt{d^2 H^4 K} + \epsilon d H^2 K))$ regret — a crucial robustness property for practical applications.

While our analysis relies on Assumption 1 to manage interobjective trade-offs, a fundamental open question is whether comparable regret bounds can be derived without this assumption. Additional directions include improving the regret bound of our algorithm by employing the policy update mechanism of He et al. (2023).

## Acknowledgements

The work described in this paper was supported by the Research Grants Council of the Hong Kong Special Administrative Region, China [GRF Project No. CityU 11215622].

## Impact Statement

This paper presents work whose goal is to advance the field of Machine Learning. There are many potential societal consequences of our work, none which we feel must be specifically highlighted here.

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

# A. Proof of Theorem 1

**Notation.** We denote $\mathrm{U}_{k,h}$, $\hat{w}_{k,h}^i$, and $\hat{Q}_{k,h}^i$ as the parameters and the estimated Q-function at episode $k$, with $\pi_k$ representing the policy at $k$-th episode. The estimated value function $\hat{V}_{k,h}^i$ is defined as $\hat{V}_{k,h}^i(x) = \hat{Q}_{k,h}^i(x, \pi_k(x, h))$. For notational simplicity, we let $\phi_{k,h} = \phi(x_{k,h}, a_{k,h})$. For any $(x, a, k, h) \in \mathcal{S} \times \mathcal{A} \times [K] \times [H]$, the surrogate Q-function for objective $i \in [m]$ is defined as

$$\bar{Q}_{k,h}^i(x,a) = r_h^i(x,a) + \int_{x' \in \mathcal{S}} \hat{V}_{k,h+1}^i(x') \mathbb{P}_h(x'|x,a) \mathrm{d}x'. \tag{9}$$

Equipped with Lemma 2 and Lemma 4, we decompose the regret for objective $i \in [m]$ as

$$
\begin{aligned}
R^i(K) &= \sum_{k=1}^K V_{\pi_*,1}^i(x_{k,1}) - V_{\pi_k,1}^i(x_{k,1}) \\
&= \sum_{k=1}^K V_{\pi_*,1}^i(x_{k,1}) - \hat{V}_{k,1}^i(x_{k,1}) + \sum_{k=1}^K \hat{V}_{k,1}^i(x_{k,1}) - V_{\pi_k,1}^i(x_{k,1}).
\end{aligned}
\tag{10}
$$

Lemma 2 establishes that

$$\hat{V}_{k,1}^i(x_{k,1}) - V_{\pi_k,1}^i(x_{k,1}) \le \sum_{h=1}^H 2\beta_k \cdot 2^{-s_{k,h}(x_{k,h})} + \sum_{h=2}^H \zeta_{k,h}^i, \tag{11}$$

while Lemma 4 implies

$$V_{\pi_*,1}^i(x_{k,1}) - \hat{V}_{k,1}^i(x_{k,1}) \le \sum_{h=1}^H 10\Lambda^i(\lambda) \cdot \beta_k \cdot 2^{-s_{k,h}(x_{k,h})}. \tag{12}$$

Substituting (11) and (12) into (10), we obtain with probability at least $1 - \delta$, for any $i \in [m]$,

$$
\begin{aligned}
R^i(K) &\le \sum_{k=1}^K \sum_{h=1}^H 12\Lambda^i(\lambda) \cdot \beta_k \cdot 2^{-s_{k,h}(x_{k,h})} + \sum_{k=1}^K \sum_{h=2}^H \zeta_{k,h}^i \\
&\le \sum_{h=1}^H \sum_{s=1}^S 12\Lambda^i(\lambda) \cdot \beta_K \cdot |\psi_{h,s}| \cdot 2^{-s} + \sum_{k=1}^K \sum_{h=2}^H \zeta_{k,h}^i,
\end{aligned}
\tag{13}
$$

where $\psi_{h,s} = \left\{ k \in [K] | 2^{-s+1} \ge \|\phi_{k,h}\|_{\mathrm{U}_h} > 2^{-s} \right\}$. The bound $S \le \log(K)$ arises from the termination threshold $1/\sqrt{K}$ in Step 7 of ALgorithm 1, as $2^{-\log(K)} \le 1/\sqrt{K}$.

By Lemma 13,

$$\sum_{\tau \in \psi_{h,s}} \|\phi_{\tau,h}\|_{\mathrm{U}_\tau} \le 5\sqrt{d|\psi_{h,s}| \log(|\psi_{h,s}|)}.$$

Since $\|\phi_{\tau,h}\|_{\mathrm{U}_\tau} > 2^{-s}$ for $\tau \in \psi_{h,s}$, this implies to

$$2^{-s}|\psi_{h,s}| \le 5\sqrt{d|\psi_{h,s}| \log(|\psi_{h,s}|)}.$$

Thus,

$$
\begin{aligned}
\sum_{s=1}^S 12\Lambda^i(\lambda) \cdot \beta_K \cdot |\psi_{h,s}| \cdot 2^{-s} &\le \sum_{s=1}^S 60\Lambda^i(\lambda) \cdot \beta_K \cdot \sqrt{d|\psi_{h,s}| \log(|\psi_{h,s}|)} \\
&\le 60\Lambda^i(\lambda) \cdot \beta_K \log(K)\sqrt{dK},
\end{aligned}
\tag{14}
$$

where the last inequality follows from the Cauchy–Schwarz inequality and $\sum_{s=1}^S |\psi_{h,s}| = K$.

For the term $\sum_{k=1}^{K} \sum_{h=2}^{H} \zeta_{k,h}^i$ in Eq. (13), observe that $\zeta_{k,h}^i = \delta_{k,h}^i - \mathbb{E}[\delta_{k,h}^i | x_{k,h}, a_{k,h}]$ forms a martingale difference sequence with $|\zeta_{k,h}^i| \leq 2H$ for all $(k,h)$. By the Azuma-Hoeffding inequality, with probability at least $1 - \delta$, for any $i \in [m]$,

$$\sum_{k=1}^{K} \sum_{h=2}^{H} \zeta_{k,h}^i \leq \sqrt{2KH^3 \cdot \log(m/\delta)}. \tag{15}$$

Combining Eq. (14) and Eq. (15) in Eq. (13), we conclude with probability at least $1 - 2\delta$, for all $i \in [m]$,

$$R^i(K) \leq 60\Lambda^i(\lambda) \cdot \beta_K \log(K)\sqrt{dH^2 K} + \sqrt{2KH^3 \cdot \log(m/\delta)}.$$

Substituting $\beta_K = 3H\sqrt{d \log(2mKH/\delta)}$, this yields

$$R^i(K) \leq \widetilde{O}\left(\Lambda^i(\lambda) \cdot \sqrt{d^2 H^4 K}\right),$$

completing the proof of Theorem 1. $\qquad\square$

## B. Proof of Proposition 1

For any objective $i \in [m]$ and timestep $h \in [H]$, the Bellman equation (1) implies:

$$\begin{aligned}
Q_{\pi,h}^i(x,a) &= r_h^i(x,a) + [\mathbb{P}_h V_{\pi,h+1}^i](x,a) \\
&= r_h^i(x,a) + \mathbb{E}_{x' \sim \mathbb{P}_h(\cdot|x,a)}[V_{\pi,h+1}^i(x')] \\
&= r_h^i(x,a) + \int_{x' \in \mathcal{S}} V_{\pi,h+1}^i(x')\mathbb{P}_h(x'|x,a)\mathrm{d}x'.
\end{aligned} \tag{16}$$

By the definition of a linear MOMDP, the reward and transition functions satisfy:

$$r_h^i(x,a) = \langle \phi(x,a), \boldsymbol{\theta}_h^i \rangle, \quad \mathbb{P}_h(x'|x,a) = \langle \phi(x,a), \boldsymbol{\mu}_h(x') \rangle.$$

Substituting these expressions into Eq. (16) yields:

$$\begin{aligned}
Q_{\pi,h}^i(x,a) &= \langle \phi(x,a), \boldsymbol{\theta}_h^i \rangle + \int_{x' \in \mathcal{S}} V_{\pi,h+1}^i(x') \cdot \langle \phi(x,a), \boldsymbol{\mu}_h(x') \rangle \mathrm{d}x' \\
&= \langle \phi(x,a), \boldsymbol{\theta}_h^i + \int_{x' \in \mathcal{S}} V_{\pi,h+1}^i(x')\boldsymbol{\mu}_h(x')\mathrm{d}x' \rangle.
\end{aligned}$$

Letting $\boldsymbol{w}_{\pi,h}^i = \boldsymbol{\theta}_h^i + \int_{x' \in \mathcal{S}} V_{\pi,h+1}^i(x')\mathrm{d}\boldsymbol{\mu}_h(x')$ finishes the proof of Proposition 1. $\qquad\square$

## C. Proof of Lemma 1

Following a similar proof of Proposition 1, we derive

$$\bar{Q}_{k,h}^i(x,a) = r_h^i(x,a) + [\mathbb{P}_h \hat{V}_{k,h+1}^i](x,a) = \langle \phi(x,a), \bar{\boldsymbol{w}}_{k,h}^i \rangle, \tag{17}$$

where $\bar{\boldsymbol{w}}_{k,h}^i = \boldsymbol{\theta}_h^i + \int_{x' \in \mathcal{S}} \hat{V}_{k,h+1}^i(x')\mathrm{d}\boldsymbol{\mu}_h(x')$. Therefore, the equation in Lemma 1 can be rewritten as

$$\langle \phi(x,a), \hat{\boldsymbol{w}}_{k,h}^i \rangle - \bar{Q}_{k,h}^i(x,a) = \langle \phi(x,a), \hat{\boldsymbol{w}}_{k,h}^i - \bar{\boldsymbol{w}}_{k,h}^i \rangle. \tag{18}$$

Recalling the expression for $\hat{\boldsymbol{w}}_{k,h}^i$ from Eq. (4), we obtain

$$\hat{\boldsymbol{w}}_{k,h}^i - \bar{\boldsymbol{w}}_{k,h}^i = \mathrm{U}_{k,h}^{-1} \sum_{\tau=1}^{k} \phi_{\tau,h} \cdot \hat{r}_{\tau,h}^i - \bar{\boldsymbol{w}}_{k,h}^i.$$

Eq. (3) indicates that $\hat{r}^i_{\tau,h} = r^i_{\tau,h} + \hat{Q}^i_{k,h+1}(x_{\tau,h+1}, a_{\tau,h+1})$ for all $\tau \in [k]$. Substituting this yields

$$\hat{\boldsymbol{w}}^i_{k,h} - \bar{\boldsymbol{w}}^i_{k,h} = \mathrm{U}^{-1}_{k,h} \left( \sum_{\tau=1}^{k} \phi_{\tau,h} \cdot \left( r^i_{\tau,h} + \hat{Q}^i_{k,h+1}(x_{\tau,h+1}, a_{\tau,h+1}) \right) - \mathrm{U}_{k,h} \bar{\boldsymbol{w}}^i_{k,h} \right)$$

$$= \mathrm{U}^{-1}_{k,h} \left( \sum_{\tau=1}^{k} \phi_{\tau,h} \cdot \left( r^i_{\tau,h} + \hat{V}^i_{k,h+1}(x_{\tau,h+1}) \right) - \mathrm{U}_{k,h} \bar{\boldsymbol{w}}^i_{k,h} \right).$$

From Eq. (17), we observe that $r^i_{\tau,h} = \langle \phi_{\tau,h}, \bar{\boldsymbol{w}}^i_{k,h} \rangle - [\mathbb{P}_h \hat{V}^i_{k,h+1}](x_{\tau,h}, a_{\tau,h})$. Substituting this into the above equation gives

$$\hat{\boldsymbol{w}}^i_{k,h} - \bar{\boldsymbol{w}}^i_{k,h} = \mathrm{U}^{-1}_{k,h} \left( \sum_{\tau=1}^{k} \phi_{\tau,h} \cdot (\langle \phi_{\tau,h}, \bar{\boldsymbol{w}}^i_{k,h} \rangle - [\mathbb{P}_h \hat{V}^i_{k,h+1}](x_{\tau,h}, a_{\tau,h}) + \hat{V}^i_{k,h+1}(x_{\tau,h+1})) - \mathrm{U}_{k,h} \bar{\boldsymbol{w}}^i_{k,h} \right)$$

$$= \mathrm{U}^{-1}_{k,h} \left( -\bar{\boldsymbol{w}}^i_{k,h} + \sum_{\tau=1}^{k} \phi_{\tau,h} \cdot \left( \hat{V}^i_{k,h+1}(x_{\tau,h+1}) - [\mathbb{P}_h \hat{V}^i_{k,h+1}](x_{\tau,h}, a_{\tau,h}) \right) \right) \tag{19}$$

where the final equality follows from $\mathrm{U}_{k,h} = \sum_{\tau=1}^{k} \phi(x_{\tau,h}, a_{\tau,h}) \phi(x_{\tau,h}, a_{\tau,h})^\top + \mathrm{I}$.

Substituting Eq. (19) into Eq. (18), we decompose the bound into two terms:

$$\left| \langle \phi(x, a), \hat{\boldsymbol{w}}^i_{k,h} - \bar{\boldsymbol{w}}^i_{k,h} \rangle \right|$$

$$\leq \underbrace{\left| \langle \phi(x, a), \mathrm{U}^{-1}_{k,h} \bar{\boldsymbol{w}}^i_{k,h} \rangle \right|}_{A_1} + \underbrace{\left| \left\langle \phi(x, a), \mathrm{U}^{-1}_{k,h} \sum_{\tau=1}^{k} \phi_{\tau,h} \cdot \left( \hat{V}^i_{k,h+1}(x_{\tau,h+1}) - [\mathbb{P}_h \hat{V}^i_{k,h+1}](x_{\tau,h}, a_{\tau,h}) \right) \right\rangle \right|}_{A_2}. \tag{20}$$

To bound $A_1$ and $A_2$, we introduce two lemmas:

**Lemma 5.** *Let $\bar{\boldsymbol{w}}^i_{k,h}$ denote the corresponding weights satisfying $\bar{Q}^i_{k,h}(x, a) = \langle \phi(x, a), \bar{\boldsymbol{w}}^i_{k,h} \rangle$ for all $(x, a, h) \in \mathcal{S} \times \mathcal{A} \times [H]$ and $i \in [m]$. Then, $\|\bar{\boldsymbol{w}}^i_{k,h}\| \leq 2H\sqrt{d}$.*

**Lemma 6.** *With probability at least $1 - \delta$, for any $(k, h) \in [K] \times [H]$ and $i \in [m]$,*

$$\left\| \sum_{\tau=1}^{k} \phi_{\tau,h} \left( \hat{V}^i_{k,h+1}(x_{\tau,h+1}) - [\mathbb{P}_h \hat{V}^i_{k,h+1}](x_{\tau,h}, a_{\tau,h}) \right) \right\|^2_{\mathrm{U}_{k,h}} \leq dH^2 \log \left( \frac{2mkH}{\delta} \right).$$

Applying Lemma 5, we bound $A_1$ as

$$A_1 = |\langle \phi(x, a), \mathrm{U}^{-1}_{k,h} \bar{\boldsymbol{w}}^i_{k,h} \rangle| \leq \|\phi(x, a)\|_{\mathrm{U}_{k,h}} \cdot \|\bar{\boldsymbol{w}}^i_{k,h}\|_{\mathrm{U}_{k,h}} \leq \|\phi(x, a)\|_{\mathrm{U}_{k,h}} \cdot \|\bar{\boldsymbol{w}}^i_{k,h}\| \leq 2H\sqrt{d} \cdot \|\phi(x, a)\|_{\mathrm{U}_{k,h}}. \tag{21}$$

For $A_2$, Lemma 6 and the Cauchy-Schwarz inequality imply

$$A_2 \leq \|\phi(x, a)\|_{\mathrm{U}_{k,h}} \cdot \left\| \sum_{\tau=1}^{k} \phi_{\tau,h} \cdot \left( \hat{V}^i_{k,h+1}(x_{\tau,h+1}) - [\mathbb{P}_h \hat{V}^i_{k,h+1}](x_{\tau,h}, a_{\tau,h}) \right) \right\|_{\mathrm{U}_{k,h}}$$

$$\leq H \sqrt{d \log \left( \frac{2mkH}{\delta} \right)} \cdot \|\phi(x, a)\|_{\mathrm{U}_{k,h}} \tag{22}$$

Taking Eq. (21) and Eq. (22) into Eq. (20), we obtain

$$|\langle \phi(x, a), \hat{\boldsymbol{w}}^i_{k,h} \rangle - \bar{Q}^i_{k,h}(x, a)| \leq \left( 2H\sqrt{d} + H \sqrt{d \log \left( \frac{2mkH}{\delta} \right)} \right) \cdot \|\phi(x, a)\|_{\mathrm{U}_{k,h}},$$

which concludes the proof of Lemma 1 since $\beta_k = 3H \sqrt{d \log \left( \frac{2mkH}{\delta} \right)}$. $\qquad \square$

# D. Proof of Lemma 2

To bound the term $\hat{V}_{k,h}^i(x_{k,h}) - V_{\pi_k,h}^i(x_{k,h})$ in Lemma 2, we begin by decomposing it as follows:

$$\hat{V}_{k,h}^i(x_{k,h}) - V_{\pi_k,h}^i(x_{k,h}) = \underbrace{\hat{V}_{k,h}^i(x_{k,h}) - \bar{Q}_{k,h}^i(x_{k,h}, \pi_k(x_{k,h}, h))}_{B_3} + \underbrace{\bar{Q}_{k,h}^i(x_{k,h}, \pi_k(x_{k,h}, h)) - V_{\pi_k,h}^i(x_{k,h})}_{B_4}. \quad (23)$$

**Analysis of $B_3$:** By Lemma 1, with probability at least $1 - \delta$, for any $(x, k, h) \in \mathcal{S} \times [K] \times [H]$ and $i \in [m]$, we have

$$\left| \langle \phi(x, \pi_k(x, h)), \hat{\boldsymbol{w}}_{k,h}^i \rangle - \bar{Q}_{k,h}^i(x, \pi_k(x, h)) \right| \leq \beta_k \| \phi(x, \pi_k(x, h)) \|_{\mathrm{U}_{k,h}}.$$

Since $\hat{V}_{k,h}^i(x) = \hat{Q}_{k,h}^i(x, \pi_k(x, h)) = \langle \phi(x, \pi_k(x, h)), \hat{\boldsymbol{w}}_{k,h}^i \rangle$, it follows that

$$\left| \hat{V}_{k,h}^i(x) - \bar{Q}_{k,h}^i(x, \pi_k(x, h)) \right| \leq \beta_k \| \phi(x, \pi_k(x, h)) \|_{\mathrm{U}_{k,h}}. \quad (24)$$

Let $s_{k,h}(x)$ denotes the stage at which Algorithm 1 selects the action for state $x \in \mathcal{S}$. From Step 10 of Algorithm 1, we have $2^{-s_{k,h}(x)} \leq \| \phi(x, \pi_k(x, h)) \|_{\mathrm{U}_{k,h}} \leq 2^{-s_{k,h}(x)+1}$. Substituting this into Eq. (24) yields

$$\left| \hat{V}_{k,h}^i(x) - \bar{Q}_{k,h}^i(x, \pi_k(x, h)) \right| \leq 2\beta_k \cdot 2^{-s_{k,h}(x)}.$$

Thus, term $B_3$ satisfies

$$B_3 = \hat{V}_{k,h}^i(x_{k,h}) - \bar{Q}_{k,h}^i(x_{k,h}, \pi_k(x_{k,h}, h)) \leq 2\beta_k \cdot 2^{-s_{k,h}(x_{k,h})}. \quad (25)$$

**Analysis of $B_4$:** Using the definition of $\bar{Q}_{k,h}^i(\cdot, \cdot)$ in Eq. (9) and the Bellman equation, we observe:

$$\bar{Q}_{k,h}^i(x, a) = r_h^i(x, a) + \int_{x' \in \mathcal{S}} \hat{V}_{k,h+1}^i(x') \mathbb{P}_h(x'|x, a) \mathrm{d}x',$$

$$V_{\pi,h}^i(x) = Q_{\pi,h}^i(x, \pi(x, h)), Q_{\pi,h}^i(x, a) = r_h^i(x, a) + \int_{x' \in \mathcal{S}} V_{\pi,h+1}^i(x') \mathbb{P}_h(x'|x, a) \mathrm{d}x'.$$

This implies $B_4$ can be expressed as

$$B_4 = \bar{Q}_{k,h}^i(x_{k,h}, \pi_k(x_{k,h}, h)) - V_{\pi_k,h}^i(x_{k,h}) = \int_{x' \in \mathcal{S}} \left( \hat{V}_{k,h+1}^i(x') - V_{k,h+1}^i(x') \right) \mathbb{P}_h(x'|x_{k,h}, \pi_k(x_{k,h}, h)) \mathrm{d}x'. \quad (26)$$

**Combining results:** Substituting (25) and (26) into (23) gives

$$\hat{V}_{k,h}^i(x_{k,h}) - V_{\pi_k,h}^i(x_{k,h}) \leq 2\beta_k \cdot 2^{-s_{k,h}(x_{k,h})} + \int_{x' \in \mathcal{S}} (\hat{V}_{k,h+1}^i(x') - V_{k,h+1}^i(x')) \mathbb{P}_h(x'|x_{k,h}, \pi_k(x_{k,h}, h)) \mathrm{d}x'.$$

Recalling the notation $\delta_{k,h}^i = \hat{V}_{k,h}^i(x_{k,h}) - V_{\pi_k,h}^i(x_{k,h})$ and $\zeta_{k,h}^i = \delta_{k,h}^i - \mathbb{E}[\delta_{k,h}^i | x_{k,h}, a_{k,h}]$ from Lemma 2, we obtain

$$\hat{V}_{k,h}^i(x_{k,h}) - V_{\pi_k,h}^i(x_{k,h}) \leq 2\beta_k \cdot 2^{-s_{k,h}(x_{k,h})} + \delta_{k,h+1}^i + \zeta_{k,h+1}^i$$

$$\leq \sum_{h'=h}^{H} 2\beta_k \cdot 2^{-s_{k,h'}(x_{k,h'})} + \sum_{h'=h+1}^{H} \zeta_{k,h'}^i,$$

where the second inequality follows by a simple inductive argument. The proof of Lemma 2 is finished. $\qquad \square$

# E. Proof of Lemma 3

We prove this lemma by mathematical induction on the objective index $i \in [m]$ to prove this lemma. First, we note that in Algorithm 2, the confidence terms of all actions are bounded by $C$. By Lemma 1, for any $(x, a) \in \mathcal{S} \times \mathcal{A}_s$ and objective $i \in [m]$, we have

$$|\hat{Q}_{k,h}^i(x, a) - \bar{Q}_{k,h}^i(x, a)| \leq \beta_k \cdot C. \quad (27)$$

**Base case** $i = 1$**:** The candidate action set $\mathcal{A}_s^0 = \mathcal{A}_s$ and the refined set $\mathcal{A}_s^1$ is constructed as

$$\mathcal{A}_s^1 = \{a \in \mathcal{A}_s^0 | \hat{Q}_h^1(x_{k,h}, \hat{a}_{k,h}^1) - \hat{Q}_h^1(x_{k,h}, a) \le 2\beta_k \cdot C\}. \tag{28}$$

Since $\pi_*(x, h) \in \mathcal{A}_s$ and $\hat{a}_{k,h}^1 \in \mathcal{A}_s^0 = \mathcal{A}_s$, Eq. (27) implies

$$\hat{Q}_{k,h}^1(x, \pi_*(x, h)) + \beta_k \cdot C \ge \bar{Q}_{k,h}^1(x, \pi_*(x, h)) \ge \bar{Q}_{k,h}^1(x, \hat{a}_{k,h}^1) \ge \hat{Q}_{k,h}^1(x, \hat{a}_{k,h}^1) - \beta_k \cdot C,$$

where the inequality $\bar{Q}_{k,h}^1(x, \pi_*(x, h)) \ge \bar{Q}_{k,h}^1(x, \hat{a}_{k,h}^1)$ follows from Assumption 1 and the optimality of $\pi_*(x, h)$. Substituting these bounds into Eq. (28), we conclude $\pi_*(x, h) \in \mathcal{A}_s^1$.

For any $a \in \mathcal{A}_s^1$, applying the confidence interval in Eq. (27) and the refined operation in Eq. (28) gives

$$\bar{Q}_{k,h}^1(x, a) \ge \hat{Q}_{k,h}^1(x, a) - \beta_k \cdot C \ge \hat{Q}_h^1(x_{k,h}, \hat{a}_{k,h}^1) - 3\beta_k \cdot C.$$

Since $\hat{a}_{k,h}^1 = \operatorname{argmax}_{a \in \mathcal{A}_s^0} \hat{Q}_{k,h}^1(x, a)$ and $\pi_*(x, h) \in \mathcal{A}_s^0$, we further derive

$$\bar{Q}_{k,h}^1(x, a) \ge \hat{Q}_h^1(x_{k,h}, \hat{a}_{k,h}^1) - 3\beta_k \cdot C \ge \hat{Q}_h^1(x_{k,h}, \pi_*(x, h)) - 3\beta_k \cdot C \ge \bar{Q}_h^1(x_{k,h}, \pi_*(x, h)) - 4\beta_k \cdot C.$$

Thus, $\bar{Q}_h^1(x_{k,h}, \pi_*(x, h)) - \bar{Q}_{k,h}^1(x, a) \le 4\beta_k \cdot C$.

**Inductive step:** Assume for any $j \in [i - 1]$ that:

$$\begin{aligned} \pi_*(x, h) &\in \mathcal{A}_s^j, \text{ and} \\ \bar{Q}_{k,h}^j(x, \pi_*(x, h)) - \bar{Q}_{k,h}^j(x, a) &\le 4\Lambda^j(\lambda) \cdot \beta_k \cdot C, \end{aligned} \tag{29}$$

then we prove the following statements hold for the objective $i$:

$$\begin{aligned} \pi_*(x, h) &\in \mathcal{A}_s^i, \text{ and} \\ \bar{Q}_{k,h}^i(x, \pi_*(x, h)) - \bar{Q}_{k,h}^i(x, a) &\le 4\Lambda^i(\lambda) \cdot \beta_k \cdot C. \end{aligned}$$

By, Assumption 1, the trade-off among different objectives are bounded by $\lambda$, which implies that

$$\bar{Q}_{k,h}^i(x, \hat{a}_{k,h}^i) - \bar{Q}_{k,h}^i(x, \pi_*(x, h)) \le \lambda \cdot \max_{j \in [i-1]} \{\bar{Q}_{k,h}^j(x, \pi_*(x, h)) - \bar{Q}_{k,h}^j(x, \hat{a}_{k,h}^j)\}.$$

Substituting Eq. (29) into the right-hand side yields

$$\bar{Q}_{k,h}^i(x, \hat{a}_{k,h}^i) - \bar{Q}_{k,h}^i(x, \pi_*(x, h)) \le \lambda \cdot 4\Lambda^{i-1}(\lambda) \cdot \beta_k \cdot C. \tag{30}$$

Applying the confidence intervals from Eq. (27) to bound $\bar{Q}_{k,h}^i(x, \hat{a}_{k,h}^i)$ and $\bar{Q}_{k,h}^i(x, \pi_*(x, h))$, we rewrite Eq. (30) as

$$\hat{Q}_{k,h}^i(x, \hat{a}_{k,h}^i) - \hat{Q}_{k,h}^i(x, \pi_*(x, h)) \le 2\beta_k \cdot C + \lambda \cdot 4\Lambda^{i-1}(\lambda) \cdot \beta_k \cdot C. \tag{31}$$

Given $\Lambda^i(\lambda) = 1 + \lambda + \cdots + \lambda^{i-1}$ for $i \in [m]$ and Step 4 of Algorithm 2, which constructs,

$$\mathcal{A}_s^i = \{a \in \mathcal{A}_s^{i-1} | \hat{Q}_h^i(x_{k,h}, \hat{a}_{k,h}^i) - \hat{Q}_h^i(x_{k,h}, a) \le (2 + 4\lambda + \cdots + 4\lambda^{i-1}) \cdot \beta_k \cdot C\},$$

Eq. (31) ensures $\pi_*(x, h) \in \mathcal{A}_s^i$.

Since $\hat{a}_{k,h}^i = \operatorname{argmax}_{a \in \mathcal{A}_s^{i-1}} \hat{Q}_h^i(x_{k,h}, a)$ and $\pi_*(x, h) \in \mathcal{A}_s^{i-1}$, the construction of $\mathcal{A}_s^i$ implies

$$\hat{Q}_h^i(x_{k,h}, \pi_*(x, h)) - \hat{Q}_h^i(x_{k,h}, a) \le \hat{Q}_h^i(x_{k,h}, \hat{a}_{k,h}^i) - \hat{Q}_h^i(x_{k,h}, a) \le (2 + 4\lambda + \cdots + 4\lambda^{i-1}) \cdot \beta_k \cdot C. \tag{32}$$

By the confidence interval in Eq. (27), we have

$$\bar{Q}_h^i(x_{k,h}, \pi_*(x, h)) \le \hat{Q}_h^i(x_{k,h}, \pi_*(x, h)) + \beta_k \cdot C, \quad \hat{Q}_h^i(x_{k,h}, a) - \bar{Q}_h^i(x_{k,h}, a) \le \beta_k \cdot C.$$

Substituting this into Eq. (32) yields

$$\bar{Q}_h^i(x_{k,h}, \pi_*(x, h)) - \bar{Q}_h^i(x_{k,h}, a) \le 4(1 + \lambda + \cdots + \lambda^{i-1}) \cdot \beta_k \cdot C.$$

This completes the proof of Lemma 3. $\qquad\square$

# F. Proof of Lemma 4

Recall the definition of $\bar{Q}^i_{k,h}(\cdot, \cdot)$ in Eq. (9) and Bellman equation:

$$\bar{Q}^i_{k,h}(x,a) = r^i_h(x,a) + \int_{x' \in \mathcal{S}} \hat{V}^i_{k,h+1}(x') \mathbb{P}_h(x'|x,a) \mathrm{d}x',$$

$$V^i_{\pi,h}(x) = Q^i_{\pi,h}(x, \pi(x,h)), Q^i_{\pi,h}(x,a) = r^i_h(x,a) + \int_{x' \in \mathcal{S}} V^i_{\pi,h+1}(x') \mathbb{P}_h(x'|x,a) \mathrm{d}x'.$$

From these, we can write $V^i_{\pi_*,h}(x)$ as

$$V^i_{\pi_*,h}(x) = \bar{Q}^i_{k,h}(x, \pi_*(x,h)) + \int_{x' \in \mathcal{S}} \left( V^i_{\pi_*,h+1}(x') - \hat{V}^i_{k,h+1}(x') \right) \mathbb{P}_h(x'|x, \pi_*(x,h)) \mathrm{d}x'.$$

Lemma 3 establishes the bound

$$\bar{Q}^i_{k,h}(x_{k,h}, \pi_*(x_{k,h},h)) - \bar{Q}^i_{k,h}(x_{k,h}, \pi_k(x_{k,h},h)) \le 8\Lambda^i(\lambda) \cdot \beta_k \cdot 2^{-s_{k,h}(x_{k,h})}$$

where $\pi_k(x_{k,h},h) \in \mathcal{A}_{s_{k,h}(x_{k,h})}$ and $\mathcal{A}_{s_{k,h}(x_{k,h})}$ is constructed by setting $C = 2^{-s_{k,h}(x_{k,h})+1}$ in the LAE procedure. Substituting this into the expression for $V^i_{\pi_*,h}(x)$, we obtain

$$\begin{aligned} V^i_{\pi_*,h}(x_{k,h}) &\le \bar{Q}^i_{k,h}(x_{k,h}, \pi_k(x_{k,h},h)) + 8\Lambda^i(\lambda) \cdot \beta_k \cdot 2^{-s_{k,h}(x_{k,h})} \\ &\quad + \int_{x' \in \mathcal{S}} \left( V^i_{\pi_*,h+1}(x') - \hat{V}^i_{k,h+1}(x') \right) \mathbb{P}_h(x'|x_{k,h}, \pi_*(x_{k,h},h)) \mathrm{d}x'. \end{aligned} \tag{33}$$

Lemma 1 provides the confidence interval

$$\begin{aligned} \bar{Q}^i_{k,h}(x_{k,h}, \pi_k(x_{k,h},h)) &\le \hat{Q}^i_{k,h}(x_{k,h}, \pi_k(x_{k,h},h)) + \beta_k \cdot \|\phi(x_{k,h}, \pi_k(x_{k,h},h))\|_{\mathrm{U}_{k,h}} \\ &\le \hat{Q}^i_{k,h}(x_{k,h}, \pi_k(x_{k,h},h)) + 2\beta_k \cdot 2^{-s_{k,h}(x_{k,h})} \\ &= \hat{V}^i_{k,h}(x_{k,h}) + 2\beta_k \cdot 2^{-s_{k,h}(x_{k,h})}. \end{aligned}$$

Substituting this inequality into (33) yields

$$\begin{aligned} V^i_{\pi_*,h}(x_{k,h}) &\le \hat{V}^i_{k,h}(x_{k,h}) + 2\beta_k \cdot 2^{-s_{k,h}(x_{k,h})} + 8\Lambda^i(\lambda) \cdot \beta_k \cdot 2^{-s_{k,h}(x_{k,h})} \\ &\quad + \max_{a \in \mathcal{A}} \int_{x' \in \mathcal{S}} \left( V^i_{\pi_*,h+1}(x') - \hat{V}^i_{k,h+1}(x') \right) \mathbb{P}_h(x'|x_{x,k}, a) \mathrm{d}x'. \end{aligned}$$

Applying induction over $h$, we have

$$V^i_{\pi_*,h}(x_{k,h}) - \hat{V}^i_{k,h}(x_{k,h}) \le \sum_{h'=h}^{H} 10\Lambda^i(\lambda) \cdot \beta_k \cdot 2^{-s_{k,h'}(x_{k,h'})}.$$

This completes the proof of Lemma 4. $\qquad\square$

# G. Proof of Theorem 2

**Notation.** Let $\mathrm{U}_{k,h}$, $\hat{w}^i_{k,h}$, and $\hat{Q}^i_{k,h}$ denote the parameters and the estimated Q-function for episode $k$, with $\pi_k$ representing the policy at $k$-th episode. The estimated value function $\hat{V}^i_{k,h}$ is defined as $\hat{V}^i_{k,h}(x) = \hat{Q}^i_{k,h}(x, \pi_k(x,h))$. For notational simplicity, we define $\phi_{k,h} = \phi(x_{k,h}, a_{k,h})$. For any $(x, a, k, h) \in \mathcal{S} \times \mathcal{A} \times [K] \times [H]$, the surrogate Q-function for objective $i \in [m]$ is given by

$$\bar{Q}^i_{k,h}(x,a) = r^i_h(x,a) + \int_{x' \in \mathcal{S}} \hat{V}^i_{k,h+1}(x') \mathbb{P}_h(x'|x,a) \mathrm{d}x'. \tag{34}$$

We now present the following proposition and lemmas, which extend key tools used in the proof of Theorem 1. Specifically, Proposition 3 extends Proposition 3, while Lemmas 7, 8, 9, and 10 extend Lemmas 1, 2, 3, and 4, respectively.

**Proposition 3.** *In an $\epsilon$-approximate linear MOMDP, for any objective $i \in [m]$ and any $k \in [K]$, there exist weights $\{\bar{\boldsymbol{w}}_{k,h}^i\}_{h \in [H]}$ such that for any $(x, a, h) \in \mathcal{S} \times \mathcal{A} \times [H]$, we have*

$$|\bar{Q}_{k,h}^i(x, a) - \langle \phi(x, a), \bar{\boldsymbol{w}}_{k,h}^i \rangle| \leq 2H\epsilon.$$

**Proof.** From the definition of $\bar{Q}_{k,h}^i(x, a)$ in Eq. (34), for all $i \in [m]$ and $h \in [H]$

$$
\begin{aligned}
\bar{Q}_{k,h}^i(x, a) &= r_h^i(x, a) + [\mathbb{P}_h \hat{V}_{k,h+1}^i](x, a) \\
&= r_h^i(x, a) + \mathbb{E}_{x' \sim \mathbb{P}_h(\cdot|x,a)}[\hat{V}_{k,h+1}^i(x')] \\
&= r_h^i(x, a) + \int_{x' \in \mathcal{S}} \hat{V}_{k,h+1}^i(x') \mathbb{P}_h(x'|x, a) \mathrm{d}x'.
\end{aligned}
\tag{35}
$$

By the definition of $\epsilon$-approximate linear MOMDP,

$$|r_h^i(x, a) - \langle \phi(x, a), \boldsymbol{\theta}_h^i \rangle| \leq \epsilon, \quad \|\mathbb{P}_h(x'|x, a) - \langle \phi(x, a), \boldsymbol{\mu}_h(x') \rangle\|_1 \leq \epsilon.$$

Taking these bounds into Eq. (35), we derive:

$$
\begin{aligned}
\bar{Q}_{k,h}^i(x, a) &\leq \langle \phi(x, a), \boldsymbol{\theta}_h^i \rangle + \epsilon + \int_{x' \in \mathcal{S}} \hat{V}_{k,h+1}^i(x') \cdot \langle \phi(x, a), \boldsymbol{\mu}_h(x') \rangle \mathrm{d}x' + H\epsilon \\
&\leq \langle \phi(x, a), \boldsymbol{\theta}_h^i + \int_{x' \in \mathcal{S}} \hat{V}_{k,h+1}^i(x') \boldsymbol{\mu}_h(x') \mathrm{d}x' \rangle + 2H\epsilon.
\end{aligned}
$$

A symmetric argument provides the lower bound:

$$\langle \phi(x, a), \boldsymbol{\theta}_h^i + \int_{x' \in \mathcal{S}} \hat{V}_{k,h+1}^i(x') \boldsymbol{\mu}_h(x') \mathrm{d}x' \rangle - 2H\epsilon \leq \bar{Q}_{k,h}^i(x, a).$$

Thus, setting $\bar{\boldsymbol{w}}_{k,h}^i = \boldsymbol{\theta}_h^i + \int_{x' \in \mathcal{S}} \hat{V}_{k,h+1}^i(x') \mathrm{d}\boldsymbol{\mu}_h(x')$ completes the proof of Proposition 3. $\square$

**Lemma 7.** *Let $\beta_k = 3H\sqrt{d \log\left(\frac{2mkH}{\delta}\right)} + 2H\epsilon\sqrt{kd}$. With probability at least $1 - \delta$, for all $(x, a, k, h) \in \mathcal{S} \times \mathcal{A} \times [K] \times [H]$ and $i \in [m]$, we have*

$$|\langle \phi(x, a), \hat{\boldsymbol{w}}_{k,h}^i \rangle - \bar{Q}_{k,h}^i(x, a)| \leq \beta_k \|\phi(x, a)\|_{\mathrm{U}_{k,h}} + 2H\epsilon.$$

**Proof.** By Proposition 3, we derive the initial bound:

$$|\langle \phi(x, a), \hat{\boldsymbol{w}}_{k,h}^i \rangle - \bar{Q}_{k,h}^i(x, a)| \leq |\langle \phi(x, a), \hat{\boldsymbol{w}}_{k,h}^i - \bar{\boldsymbol{w}}_{k,h}^i \rangle| + 2H\epsilon. \tag{36}$$

From the definition of $\hat{\boldsymbol{w}}_{k,h}^i$ in Eq. (4), we express the difference:

$$\hat{\boldsymbol{w}}_{k,h}^i - \bar{\boldsymbol{w}}_{k,h}^i = \mathrm{U}_{k,h}^{-1} \sum_{\tau=1}^k \phi_{\tau,h} \cdot \hat{r}_{\tau,h}^i - \bar{\boldsymbol{w}}_{k,h}^i.$$

Using the surrogate reward relation in Eq. (3), where $\hat{r}_{\tau,h}^i = r_{\tau,h}^i + \hat{Q}_{k,h+1}^i(x_{\tau,h+1}, a_{\tau,h+1})$ for all $\tau \in [k]$, we expand:

$$
\begin{aligned}
\hat{\boldsymbol{w}}_{k,h}^i - \bar{\boldsymbol{w}}_{k,h}^i &= \mathrm{U}_{k,h}^{-1} \left( \sum_{\tau=1}^k \phi_{\tau,h} \cdot \left( r_{\tau,h}^i + \hat{Q}_{k,h+1}^i(x_{\tau,h+1}, a_{\tau,h+1}) \right) - \mathrm{U}_{k,h} \bar{\boldsymbol{w}}_{k,h}^i \right) \\
&= \mathrm{U}_{k,h}^{-1} \left( \sum_{\tau=1}^k \phi_{\tau,h} \cdot \left( r_{\tau,h}^i + \hat{V}_{k,h+1}^i(x_{\tau,h+1}) \right) - \mathrm{U}_{k,h} \bar{\boldsymbol{w}}_{k,h}^i \right).
\end{aligned}
$$

Let $\tilde{\mathbb{P}}_h(\cdot|x, a) = \langle \phi(x, a), \boldsymbol{\mu}_h(\cdot) \rangle$. Recalling the proof of Proposition 3 that $\bar{\boldsymbol{w}}_{k,h}^i = \boldsymbol{\theta}_h^i + \int_{x' \in \mathcal{S}} \hat{V}_{k,h+1}^i(x') \mathrm{d}\boldsymbol{\mu}_h(x')$, we restructure the above equation to

$$
\begin{aligned}
\hat{\boldsymbol{w}}_{k,h}^i - \bar{\boldsymbol{w}}_{k,h}^i = &-\mathrm{U}_{k,h}^{-1} \bar{\boldsymbol{w}}_{k,h}^i + \mathrm{U}_{k,h}^{-1} \sum_{\tau=1}^k \phi_{\tau,h} \left( r_{\tau,h}^i - \langle \phi_{\tau,h}, \boldsymbol{\theta}_h^i \rangle + [(\mathbb{P}_h - \tilde{\mathbb{P}}_h) \hat{V}_{k,h+1}^i](x_{\tau,h}, a_{\tau,h}) \right) \\
&+ \mathrm{U}_{k,h}^{-1} \left( \sum_{\tau=1}^k \phi_{\tau,h} \cdot \left( \hat{V}_{k,h+1}^i(x_{\tau,h+1}) - [\mathbb{P}_h \hat{V}_{k,h+1}^i](x_{\tau,h}, a_{\tau,h}) \right) \right).
\end{aligned}
\tag{37}
$$

Taking the Eq. (37) into Eq. (36), we have

$$
\begin{aligned}
&\left| \left\langle \phi(x,a), \hat{\boldsymbol{w}}_{k,h}^i - \bar{\boldsymbol{w}}_{k,h}^i \right\rangle \right| \\
&\leq \underbrace{\left| \left\langle \phi(x,a), \mathrm{U}_{k,h}^{-1} \bar{\boldsymbol{w}}_{k,h}^i \right\rangle \right|}_{A_1} + \underbrace{\left| \left\langle \phi(x,a), \mathrm{U}_{k,h}^{-1} \sum_{\tau=1}^{k} \phi_{\tau,h} \cdot \left( \hat{V}_{k,h+1}^i(x_{\tau,h+1}) - [\mathbb{P}_h \hat{V}_{k,h+1}^i](x_{\tau,h}, a_{\tau,h}) \right) \right\rangle \right|}_{A_2} \\
&+ \underbrace{\left| \left\langle \phi(x,a), \mathrm{U}_{k,h}^{-1} \sum_{\tau=1}^{k} \phi_{\tau,h} \left( r_{\tau,h}^i - \langle \phi_{\tau,h}, \boldsymbol{\theta}_h^i \rangle + (\mathbb{P}_h - \tilde{\mathbb{P}}_h) \hat{V}_{k,h+1}^i(x_{\tau,h}, a_{\tau,h}) \right) \right\rangle \right|}_{A_3}.
\end{aligned}
\tag{38}
$$

Term $A_1$ and $A_2$ are bounded by the same methodology as Lemma 1. For $A_3$, noting that $|r_{\tau,h}^i - \langle \phi_{\tau,h}, \boldsymbol{\theta}_h^i \rangle| \leq \epsilon$ and $\|\mathbb{P}_h - \tilde{\mathbb{P}}_h\|_1 \leq \epsilon$, we can bound $A_3$ as

$$
\begin{aligned}
A_3 &\leq 2H\epsilon \cdot \left| \sum_{\tau=1}^{k} \phi(x,a)^\top \mathrm{U}_{k,h}^{-1} \phi_{\tau,h} \right| \\
&\leq 2H\epsilon \cdot \sqrt{ \sum_{\tau=1}^{k} \phi(x,a)^\top \mathrm{U}_{k,h}^{-1} \phi(x,a) \cdot \sum_{\tau=1}^{k} \phi_{\tau,h}^\top \mathrm{U}_{k,h}^{-1} \phi_{\tau,h} } \\
&\leq 2H\epsilon \sqrt{kd} \cdot \|\phi(x,a)\|_{\mathrm{U}_{k,h}}.
\end{aligned}
\tag{39}
$$

Integrating Eqs. (21), Eq. (22), and Eq. (39) into Eq. (38) produces:

$$
|\langle \phi(x,a), \hat{\boldsymbol{w}}_{k,h}^i \rangle - \bar{Q}_{k,h}^i(x,a)| \leq \left( 3H\sqrt{d \log\left( \frac{2mkH}{\delta} \right)} + 2H\epsilon\sqrt{kd} \right) \cdot \|\phi(x,a)\|_{\mathrm{U}_{k,h}} + 2H\epsilon,
$$

which concludes the proof of Lemma 7. $\qquad \square$

The proofs of the following three lemmas follow analogous arguments to their counterparts in Lemmas 2, 3, and 4, requiring only the replacement of $\beta_k \|\phi(x,a)\|_{\mathrm{U}_{k,h}}$ with $\beta_k \|\phi(x,a)\|_{\mathrm{U}_{k,h}} + 2H\epsilon$. We therefore omit their full proofs for brevity.

**Lemma 8.** *Let $\delta_{k,h}^i = \hat{V}_{k,h}^i(x_{k,h}) - V_{\pi_k,h}^i(x_{k,h})$ and $\zeta_{k,h}^i = \delta_{k,h}^i - \mathbb{E}[\delta_{k,h}^i | x_{k,h}, a_{k,h}]$. With probability at least $1-\delta$, for any $(k,h) \in [K] \times [H]$ and $i \in [m]$, we have*

$$
\hat{V}_{k,h}^i(x_{k,h}) - V_{\pi_k,h}^i(x_{k,h}) \leq \sum_{h'=h}^{H} 2\beta_k \cdot 2^{-s_{k,h'}(x_{k,h'})} + \sum_{h'=h+1}^{H} \zeta_{k,h'}^i + 2H(H - h + 1)\epsilon,
$$

*where $s_{k,h'}(x_{k,h'})$ denotes the stage at which LLRL selects the action for state $x_{k,h'}$.*

**Lemma 9.** *For Algorithm 2, if $\pi_*(x,h) \in \mathcal{A}_s$, then with probability at least $1-\delta$, for any objective $i \in [m]$ and action $a \in \mathcal{A}_s^i$, we have*

$$
\pi_*(x,h) \in \mathcal{A}_s^i, \text{ and}
$$
$$
\bar{Q}_{k,h}^i(x, \pi_*(x,h)) - \bar{Q}_{k,h}^i(x,a) \leq 4\Lambda^i(\lambda) \cdot \beta_k \cdot (C + 2H\epsilon).
$$

**Lemma 10.** *With probability at least $1-\delta$, for all $(x,a,k,h) \in \mathcal{S} \times \mathcal{A} \times [K] \times [H]$ and $i \in [m]$, we have*

$$
V_{\pi_*,h}^i(x_{k,h}) - \hat{V}_{k,h}^i(x_{k,h}) \leq \sum_{h'=h}^{H} 10\Lambda^i(\lambda) \cdot \beta_k \cdot 2^{-s_{k,h'}(x_{k,h'})} + 20\Lambda^i(\lambda) \cdot (H - h + 1)H\epsilon
$$

*where $s_{k,h'}(x_{k,h'})$ denotes the stage at which LLRL selects the action for state $x_{k,h'}$.*

**Proof of Theorem 2:** Using Lemma 8 and Lemma 10, we decompose the regret for each objective $i \in [m]$ as:

$$R^i(K) = \sum_{k=1}^{K} V_{\pi_*,1}^i(x_{k,1}) - V_{\pi_k,1}^i(x_{k,1})$$
$$= \sum_{k=1}^{K} V_{\pi_*,1}^i(x_{k,1}) - \hat{V}_{k,1}^i(x_{k,1}) + \sum_{k=1}^{K} \hat{V}_{k,1}^i(x_{k,1}) - V_{\pi_k,1}^i(x_{k,1}).$$

Following an analogous procedure to Theorem 1, we derive:

$$R^i(K) \leq 60\Lambda^i(\lambda) \cdot \beta_K \log(K)\sqrt{dH^2 K} + \sqrt{2KH^3 \cdot \log(m/\delta)} + 22\Lambda^i(\lambda) \cdot H^2 K\epsilon.$$

Substituting $\beta_K = 3H\sqrt{d\log(2mKH/\delta)} + 2H\epsilon\sqrt{Kd}$ yields:

$$R^i(K) \leq \tilde{O}\left(\Lambda^i(\lambda) \cdot \left(\sqrt{d^2 H^4 K} + dH^2 K\epsilon\right)\right).$$

This completes the proof of Theorem 2. $\qquad\square$

## H. Proof of Lemma 5

By the Bellman equation in Eq. (1), for any $(x, a, k, h) \in \mathcal{S} \times \mathcal{A} \times [K] \times [H]$ and $i \in [m]$, we have

$$\bar{Q}_{k,h}^i(x, a) = r_h^i(x, a) + [\mathbb{P}_h \hat{V}_{k,h+1}^i](x, a).$$

Applying reasoning analogous to Proposition 1, we derive

$$\bar{w}_{k,h}^i = \theta_h^i + \int_{x' \in \mathcal{S}} \hat{V}_{k,h+1}^i(x')\mathrm{d}\mu_h(x').$$

Given the bounded reward $r_h^i(x, a) \in [0, 1]$, the value function satisfies $\hat{V}_{k,h+1}^i(x') \leq H$ for any state $x'$. Therefore,

$$\|\theta_h\| \leq \sqrt{d}, \text{ and } \left\|\int_{x' \in \mathcal{S}} \hat{V}_{k,h+1}^i(x')\mathrm{d}\mu_h(x')\right\| \leq H\sqrt{d},$$

which finishes the proof. $\qquad\square$

## I. Proof of Lemma 6

According to Lemma 11, for any fixed $i \in [m]$ and $h \in [H]$, with probability at least $1 - \delta$, the following inequality holds for all $k \geq 1$:

$$\left\|\sum_{\tau=1}^{k} \phi_{\tau,h}\left[\hat{V}_{k,h+1}^i(x_{\tau,h+1}) - [\mathbb{P}_h \hat{V}_{k,h+1}^i](x_{\tau,h}, a_{\tau,h})\right]\right\|_{\mathrm{U}_{k,h}}^2 \leq 2H^2 \log\left[\frac{\sqrt{\det(\mathrm{U}_k)}}{\delta}\right].$$

Lemma 12 simplifies this bound via the inequality det, yielding:

$$\left\|\sum_{\tau=1}^{k} \phi_{\tau,h}\left[\hat{V}_{k,h+1}^i(x_{\tau,h+1}) - [\mathbb{P}_h \hat{V}_{k,h+1}^i](x_{\tau,h}, a_{\tau,h})\right]\right\|_{\mathrm{U}_{k,h}}^2 \leq 2H^2 \log\left[\frac{\sqrt{(1+k/d)^d}}{\delta}\right] \leq dH^2 \log\left[\frac{1+k}{\delta}\right].$$

Applying a union bound over all $i \in [m]$ and $h \in [H]$ completes the proof. $\qquad\square$

## J. Auxiliary Lemmas

**Lemma 11.** *(Abbasi-yadkori et al., 2011, Theorem 1) Let $\{\epsilon_t\}_{t=1}^{T}$ be a real-valued stochastic process adapted to filtration $\{\mathcal{F}_t\}_{t=0}^{\infty}$, where $\epsilon_t | \mathcal{F}_{t-1}$ be zero-mean and $\sigma$-subGaussian, i.e.,*

$$\mathbb{E}[\epsilon_t | \mathcal{F}_{t-1}] = 0, \text{ and } \forall \lambda \in \mathbb{R}, \mathbb{E}[e^{\lambda \epsilon_t} | \mathcal{F}_{t-1}] \leq e^{\lambda^2 \sigma^2 / 2}.$$

*Let $\{\phi_t\}_{t=0}^{\infty} \subseteq \mathbb{R}^d$ be a stochastic process where $\phi_t \in \mathcal{F}_{t-1}$. Assume $\mathrm{U}_0$ is a $d \times d$ positive definite matrix, and let $\mathrm{U}_t = \mathrm{U}_0 + \sum_{s=1}^{t} \phi_s \phi_s^{\top}$. Then for any $\delta > 0$, with probability at least $1 - \delta$, for all $t \geq 0$:*

$$\left\| \sum_{s=1}^{t} \phi_s \epsilon_s \right\|_{\mathrm{U}_t}^2 \leq 2\sigma^2 \log \left( \frac{\sqrt{\det(\mathrm{U}_t)/\det(\mathrm{U}_0)}}{\delta} \right).$$

**Lemma 12.** *(Abbasi-yadkori et al., 2011, Lemma 10) Let $\phi_1, \phi_2, \ldots, \phi_t \in \mathbb{R}^d$ satisfy $\|\phi_s\|_2 \leq L$ for any $1 \leq s \leq t$. Define $\mathrm{U}_t = \lambda \mathrm{I} + \sum_{s=1}^{t} \phi_s \phi_s^{\top}$ for some $\lambda > 0$. Then:*

$$\det(\mathrm{U}_t) \leq (\lambda + tL^2/d)^d.$$

**Lemma 13.** *(Chu et al., 2011, Lemma 3) Let $\phi_1, \phi_2, \ldots, \phi_t \in \mathbb{R}^d$ and $\mathrm{U}_t = \mathrm{I} + \sum_{s=1}^{t} \phi_s \phi_s^{\top}$. Then:*

$$\sum_{s=1}^{t} \|\phi_s\|_{\mathrm{U}_s} \leq 5\sqrt{dt \log t}.$$

