# OpenReview forum: "Multi-objective Linear Reinforcement Learning with Lexicographic Rewards"
_ICML.cc/2025/Conference — ICML 2025 poster_

### Official Review · Reviewer_xDG7 · 2025-03-06

**Overall Recommendation:** 3

**Summary:**

This work focuses on development of an algorithmic framework with theoretical performance guarantees in Multi-objective RL where the underlying Multi-Objective Markov Decision Process (MO-MDP) is assumed to be linear. The algorithmic strategy optimizes for lexicographic rewards which are essentially hierarchically ordered. To this end, Lexicographic Linear RL (LLRL), is proposed in the finite-horizon episodic learning setup. The method refines agent's policy over time by backward pass updates on the model parameters, careful management of exploration-exploitation trade-off, and handling lexicographic rewards through multi-stage action refinement via a dedicated action elimination routine. The paper also surfaces key challenges with MORL and connects them to its key algorithmic innovation pieces.Finally, a mathematical analysis of LLRL's performance measured in regret against optimal rewards is also presented.

**Claims And Evidence:**

The authors' central claim is around developing an efficient strategy for MOLRL with lexicographic rewards settings. To this end, LLRL is presented, and the evidence of LLRL's performance is captured in theoretical regret analysis (Theorem 1, 2). To the best of my knowledge, an empirical performance evaluation is missing for LLRL.

**Essential References Not Discussed:**

I think the paper provides a good overview of the relevant literature.

**Experimental Designs Or Analyses:**

A thorough experimental analysis of LLRL is missing.

**Methods And Evaluation Criteria:**

The method LLRL is proposed towards solving an MOMDP in lexicographic rewards setting, and it is being benchmarked by worst case regret bound against a optimal policy. To the best of my understanding, both the method and the evaluation criteria is relevant to the underlying problem setup.

**Other Comments Or Suggestions:**

1. While the paper is well-written overall, some of the notation can be dense and difficult to follow. Consider simplifying the notation where possible and providing more intuitive explanations of the key concepts. A diagramatic workflow could be helpful in this regard.

2. Investigate alternative assumptions for managing inter-objective trade-offs and explore whether comparable regret bounds can be derived without Assumption 1.

**Other Strengths And Weaknesses:**

Strengths:
1. The LLRL algorithm is well-described with the key techniques and challenges being clearly highlighted.
2. The proofs are rigorous and detailed.
3. The results in the misspecified setting are interesting and valuable.


Weakeness:
1. The lack of experimental results is a significant limitation.

2. The assumption that the transition kernel and reward function are linear may limit the applicability of the algorithm in practice.

**Questions For Authors:**

1. How scalable will be LLRL ? In other words, what are the computational complexities of the LLRL algorithm and the LAE procedure?

2. Can we have a mathematical sketch of how the performance would look like in the absense of Assumption 1 ? Also, is there any other way based on practical considerations that we can measure these objective trade-offs ?

**Relation To Broader Scientific Literature:**

In my understanding, the paper discusses relevant prior works on linear RL as well as MORL, identifies a concrete gap for MOMDP with lexicographic reward structure. The proposed LLRL method addresses the aforementioned problem statement, and is supported by Theorem 1,2 where the regret bounds are derived for a finite horizon setup. However, an empirical validation of LLRL is missing.

**Theoretical Claims:**

The key theoretical contributions are stated in Theorem 1,2 providing LLRL's regret bounds. These results are then proven using intermediate supporting results through Appendix A-J. In my understanding, the results are derived systematically using well-known statistical concentration inequalities, and linear algebra results.

---

> ### Author Rebuttal · Authors · 2025-03-31
>
> We sincerely appreciate your constructive feedback. We have carefully considered your concerns (including Weaknesses and Questions raised), and our responses are provided below.
>
> ---
>
> *W1. The lack of experimental results is a significant limitation.*
>
> Thank you for raising this issue. The absence of empirical validation in our work aligns with the foundational single-objective linear MDP studies (Jin et al. 2020; Zanette et al. 2020; He et al. 2023), which similarly omit experiments due to challenges in constructing valid linear MDP benchmarks (e.g., enforcing low-rank dynamics and linear payoff structures). We plan to address this limitation in two phases: **i)** Synthetic experiments will be added to illustrate key theoretical properties. **ii)** Comprehensive empirical comparisons against heuristic baselines and ablations will be conducted in follow-up work.
>
> ---
>
> *W2. The assumption that the transition kernel and reward function are linear may limit the applicability of the algorithm in practice.*
>
> We acknowledge that the linear assumptions may not hold universally in all practical scenarios, particularly in environments with highly complex dynamics or non-linear relationships. However, linearity allows us to leverage well-established mathematical tools from linear algebra and convex optimization, which are essential for deriving rigorous regret bounds. Similar assumptions are common in foundational RL theory (e.g., Jin et al. (2020), Zanette et al. (2020), and He et al. (2023)) to balance generality and analyzability.
>
> Moreover, in finite state-action spaces, any nonlinear system can be represented as a linear MDP by encoding each state-action pair $(x, a)$ as a one-hot feature vector in $\mathbb{R}^d$, where $d=|\mathcal{S}| \times |\mathcal{A}|$. The transition kernel $\mathbb{P}(x' \mid x, a)$ and reward function $r(x, a)$ can be expressed as inner products between the feature vector of $(x, a)$ and learnable parameters.
>
> In the future, we will try to adopt techniques of generalized linear bandits or Lipschitz bandits to extend the model into generalized linear or Lipschitz.
>
> ---
>
> *Q1. How scalable will be LLRL? In other words, what are the computational complexities of the LLRL algorithm and the LAE procedure?*
>
> The computational cost of LLRL mainly lies on LAE and policy update (Steps 19-24). Below, we provide a detailed complexity analysis of Algorithm 1:
>
> 1. Step 7: The complexity is $O(d^2|\mathcal{A}|)$.
>
> 2. Step 8: LAE requires $O(md|\mathcal{A}|)$ computations.
>
> 3. Step 20: The complexity is $O(d^2)$, as $U_h$ can be updated incrementally.
>
> 4. Step 21: The complexity is $O(mk)$ for updating $m \cdot k$ values.
>
> 5. Step 22: The complexity is $O(mkd+md^2)$, dominated by inverting $U_h$ ($O(d^2)$) and computing $m$ linear regressions ($O(mkd+md^2)$).
>
> 6. Step 23: No additional resources are required, as Q-values can be updated directly from retained $\{\hat{w}_h^i\}_{i\in[m]}$.
>
> - Summing across all $H$ MDP layers, the complexity of $k$-th round is $O(Hmd|\mathcal{A}|+Hd^2|\mathcal{A}|+Hmkd+Hmd^2)$.
> - Summing over $K$ rounds, the **overall computational complexity of LLRL** is $O(KHd|\mathcal{A}|(m+d)+KHmd(K+d))$.
>
> We will incorporate this complexity analysis in the revised paper to clarify scalability.
>
> ---
>
> *Q2. Can we have a mathematical sketch of how the performance would look like in the absense of Assumption 1? Also, is there any other way based on practical considerations that we can measure these objective trade-offs?*
>
> In the context of lexicographic bandit problems, Huyuk and Tekin [1] establish regret bounds without relying on assumptions analogous to Assumption 1 in our work. Extending their analysis to linear MDPs, we hypothesize that a similar regret bound of order $O((d^2H^4K)^{\frac{2}{3}})$ may hold, though a formal proof remains an open question for future investigation.
>
> Regarding practical trade-off quantification, domain-specific expertise often provides empirically grounded ratios for prioritizing objectives. For example, environmental policy frameworks frequently employ comparative metrics where 1 tonne of SO2 emissions is considered $\leq10$ times as harmful as 1 tonne of CO emissions (Podinovski [2], 1999), which directly corresponds to $\lambda=10$.
>
> *[1] Huyuk, A. and Tekin, C. Multi-objective multi-armed bandit with lexicographically ordered and satisficing objectives. Machine Learning, 110(6):1233–1266, 2021.*
>
> *[2] Podinovski, V. V. A dss for multiple criteria decision analysis with imprecisely specified trade-offs. European Journal of Operational Research, 113(2):261–270, 1999.*
>
> ---
>
> **Other Comments Or Suggestions:** *Notations and Alternative Assumptions.*
>
> Thank you for your positive feedback and constructive suggestions. In the revised paper, we will carefully revise all notation and compile it into a summary table for clarity. Meanwhile, we will try to establish comparable regret bounds without Assumption 1 in the future work.

---

> > ### Comment · Reviewer_xDG7 · 2025-04-05
> >
> > I would like to thank the authors for providing the clarifications and revision plan for some of the comments. It appears that authors have acknowledged some of the current highlighted limitations which deserves careful considerations, for instance, related to experiments, and assumptions. Therefore, I would like to maintain my original score.

---

### Official Review · Reviewer_6izD · 2025-03-12

**Overall Recommendation:** 2

**Summary:**

This paper studies multi-objective RL (MORL) with lexicographic rewards in linear MDPs, where rewards comprise hierarchically ordered objectives. A key challenge in MORL is the failure of Bellman optimality. They propose the LLRL algorithm and establish the first regret bound for MORL under a certain assumption (Assumption 1).

**Claims And Evidence:**

Yes.

**Essential References Not Discussed:**

No.

**Experimental Designs Or Analyses:**

N/A

**Methods And Evaluation Criteria:**

N/A

**Other Comments Or Suggestions:**

Typo: The inequality under Assumption 1: $\text{LHS} \le \lambda \cdot \max_{j \in [i - 1]} \\{ r_h^j(x, a_2) - r_h^j (x, a_1) \\}$

**Other Strengths And Weaknesses:**

**Strength:** This paper establishes the first theoretical regret bound for MORL.

**Weakness:** See Questions.

**Questions For Authors:**

1. What is the definition of "$a_2$ lexicographically dominates $a_1$" in Assumption 1? Does it mean the reward vector $[r_h^1(x,a_2), \cdots, r_h^m(x,a_2)]$ lexicographically dominates vector $[r_h^1(x,a_1), \cdots, r_h^m(x,a_1)]$?
2. Following Q1, I have one question regarding the proof of Lemma 3 in Appendix E. Specifically, why would Assumption 1 result in the argument in lines 797-800?

**Relation To Broader Scientific Literature:**

N/A

**Theoretical Claims:**

My major concern is on Assumption 1. I am unable to find the definition of "$a_2$ lexicographically dominates $a_1$" in the main text (please point it out if there is any). Please refer to for follow-up questions.

---

> ### Author Rebuttal · Authors · 2025-03-31
>
> Many thanks for your constructive reviews. We have carefully considered your concerns and our responses are provided as follows.
>
> ---
>
> *Q1. What is the definition of $a_2$ lexicographically dominates $a_1$ in Assumption 1? Does it mean the reward vector $[r_h^1(x, a_2),\cdots,r_h^m(x, a_2)]$ lexicographically dominates vector $[r_h^1(x, a_1),\cdots,r_h^m(x, a_1)]$?*
>
> Yes, the definition aligns with your interpretation: $a_2$ lexicographically dominates $a_1$ if and only if the reward vector $[r_h^1(x, a_2),\cdots,r_h^m(x, a_2)]$ lexicographically dominates vector $[r_h^1(x, a_1),\cdots,r_h^m(x, a_1)]$.
>
> ---
>
> *Q2. Following Q1, I have one question regarding the proof of Lemma 3 in Appendix E. Specifically, why would Assumption 1 result in the argument in lines 797-800?*
>
> We appreciate this critical observation. Upon re-examination, we recognize that Assumption 1 alone does not suffice to support the argument in lines 797-800. This oversight has led us to propose a revised assumption, detailed below.
>
> **Key Revision Rationale:** Initially, we presumed that if individual rewards satisfied Assumption 1 (lexicographic dominance on immediate rewards), their weighted aggregation $\bar{Q}\_{k,h}^i$ would inherently preserve this property. However, this reasoning neglected the temporal dynamics of MDPs: actions at step $h$ influence not only immediate rewards but also future state distributions (via $\mathbb{P}\_h$). The original assumption only bounded trade-offs in immediate rewards ($r\_h^i$), failing to account for long-term value interactions in $\bar{Q}\_{k,h}^i$.
>
> **Revised Assumption:** Let $\tilde{Q}^i\_h(x,a)=r\_{h}^i(x,a)+\[\mathbb{P}\_h \tilde{V}^i\_{h+1}\](x,a)$ for any $i\in[m]$ and $(x,a,h)\in S\times A\times [H]$. Let $\pi_*(x,h)$ denote the action chosen by the lexicographically optimal policy at $(x,h)$.  We assume the trade-off among objectives is governed by $\lambda\geq0$, such that for all $h\in[H]$ and $i\in[m]$,
> $$
> \tilde{Q}^i\_{h}(x,a)-\tilde{Q}^i\_{h}(x,\pi_*(x,h))\leq \lambda \cdot \max\_{j\in[i-1]}\left\\{\tilde{Q}^i\_{h}(x,\pi\_*(x,h))-\tilde{Q}^i\_{h}(x,a)\right\\}.
> $$
> Here, $\tilde{V}^i_h(x)=\langle w(x), \mathbf{r}^i_{h:H}\rangle$, where $w(x)\in\mathbb{R}^{H-h+1}$ is a shared weighting vector across all objectives, and $\mathbf{r}^i_{h:H}=[r^i_{h}(\cdot,\cdot),r^i_{h+1}(\cdot,\cdot),\cdots, r^i_{H}(\cdot,\cdot)]$.
>
> **Lines 797-800:** Under the revised assumption, we can demonstrate that Lines 797-800 hold because the action-value function $\bar{Q}^i_h(x,a)$ decomposes as $r_{h}^i(x,a) + \[\mathbb{P}\_h \hat{V}^i_{k,h+1}\](x,a)$, where $\hat{V}^i_{k,h+1}$ represents a weighted sum of rewards from step $h+1$ to $H$. Crucially, the weighting parameter $w(x) = \phi(x, \pi_k(x,h))^\top U_h^{-1}F_h$ is shared across all objectives, ensuring consistency in multi-objective optimization. Here, $F_h \in \mathbb{R}^{d\times k}$ denotes the feature matrix comprising historical state-action pairs, defined as $F_h = [\phi(x_{1,h},a_{1,h}), \ldots, \phi(x_{k,h},a_{k,h})]$.
>
> **Comparison with Assumption 1:**
>
> - *Advantage.* Assumption 1 requires that for any action $a_1,a_2\in A$, if $a_2$ lexicographically dominates $a_1$, then their rewards satisfies the trade-off value $\lambda$. The revised assumption fixed the one action as $\pi_*(x,h)$, relaxing the trade-off among actions.
> - *Limitation.* The revised assumption introduces a shared weight $w(x)$ across objectives, ensuring **consistent reward processing** via $\tilde{V}^i_{h+1}(x)$. While this is natural in tabular MDPs (where $w(x)$ represents visit-count normalization), it imposes stricter conditions in linear MDPs due to the non-fixed nature of $w(x)$.
>
> ---
>
> We thank the reviewer for prompting this clarification, which strengthens our theoretical framework. We are happy to answer more questions.

---

> > ### Comment · Reviewer_6izD · 2025-04-02
> >
> > Thank you for the detailed explanation. This addresses my problem with Lemma 3. However, under the revised assumption, it seems that the lexicographically optimal policy can be directly solved by standard single-objective RL, where the rewards given by $R_h(x,a) := \sum_{i=1}^m \lambda^{t(i-1)} \cdot r_h^i(x, a)$ for some integer $t := t(\lambda)$. This suggests the revision might be too restrictive. Below, I analyze the state-less case (but I think it can be generalized directly to the episodic setting).
> >
> > Consider that $a^\star$ lexicographically dominates $a$. We assume $r^1(a^\star) > r^1(a)$ WLOG. It can be shown that $q(a^\star) \ge q(a)$, where $q(a) := \sum_{i=1}^m \lambda^{t(i-1)} \cdot r^i(a)$, for some integer $t$. First, we select $i_1 := \arg\max_{j \in [m]} r^j(a^\star) - r^j(a)$ and the "tail summation" over $j = i_1, i_1 + 1, \cdots, m$ satisfies $$\sum_{j=i_1}^m \lambda^{t(j-1)} \cdot (r^j(a^\star) - r^j(a)) \ge (1 - \sum_{j=i_1+1}^m \lambda^{t(j-i_1)+1}) \cdot \lambda^{t(i_1 - 1)}(r^{i_1}(a^\star) - r^{i_1}(a)).$$ Choose $t$ such that $\lambda^{t+1} \le 1 - \lambda^t$, and the above difference is positive. Next, we select $i_2 := \arg\max_{j \in [i_1 - 1] } r^j(a^\star) - r^j(a)$ and analyze the weighted summation over $j = i_2, i_2 + 1, \cdots, i_1 - 1$. Repeat the above procedure until $i_n=1$, and the argument is proved.

---

> > > ### Author Response · Authors · 2025-04-02
> > >
> > > Thank you for your response. I am unclear about your strategy for arm selection. Do you propose selecting arm $a_t$ as follows?
> > >
> > > $$
> > > a_t = \text{argmax}\_{a \in \mathcal{A}} \sum_{i=1}^m \lambda^{m-i} \hat{Q}_h^i(x, a)
> > > $$
> > > If this is your basic idea, I identify two potential concerns:
> > >
> > > 1. **Case when $\lambda = 0$:** In this case, this strategy becomes invalid because $\sum_{i=1}^m \lambda^{m-i} \hat{Q}_h^i(x, a) =  \hat{Q}_h^m(x, a) $ for any arm $a \in \mathcal{A}$.
> > > 2. **Case when $\lambda > 0$:** In this case, this strategy introduces additional regret due to the weighted aggregation of rewards. We provide a detailed analysis below.
> > >
> > > Since rewards are aggregated via a weighted sum, we must analyze the regret of this weighted sum across objectives:
> > > $$
> > > \lambda^{m-1} R^1(K) + \lambda^{m-2} R^2(K) + \ldots + R^m(K) = \sum_{\tau=1}^K \sum_{i=1}^m \lambda^{m-i} \left( V_{\pi_*,1}^i(x_{k,1}) - V_{\pi_k,1}^i(x_{k,1}) \right)= \sum_{\tau=1}^K \left[ \sum_{i=1}^m \lambda^{m-i} V_{\pi_*,1}^i(x_{k,1}) \right] - \left[ \sum_{i=1}^m \lambda^{m-i} V_{\pi_k,1}^i(x_{k,1}) \right].
> > > $$
> > > Following an analysis similar to that of single-objective MOMDP (Jin et al., 2020), the regret bound for the weighted sum method is:
> > >
> > > $$
> > > \lambda^{m-1} R^1(K) + \lambda^{m-2} R^2(K) + \ldots + R^m(K) \leq \sum_{i=1}^m \lambda^{m-i} \widetilde{O}\left( \sqrt{d^2 H^4 K} \right).
> > > $$
> > > This result shows that when using a weighted sum of rewards, **the regret bounds of all objectives are scaled with $m$**. For instance, the regret bound for the most important objective $(i = 1)$ becomes:
> > >
> > > $$
> > > R^1(K) \leq \sum_{i=1}^m \lambda^{1-i} \widetilde{O}\left( \sqrt{d^2 H^4 K} \right),
> > > $$
> > > which depends on both $\lambda$ and the number of objectives $m$. In contrast, our algorithm achieves a regret bound of $\widetilde{O}(\sqrt{d^2 H^4 K})$ for the first objective, **independent of $m$**.
> > >
> > > ---
> > >
> > > Thank you again for your response. We are happy to answer more questions.

---

### Official Review · Reviewer_8VYt · 2025-03-12

**Overall Recommendation:** 4

**Summary:**

This paper studies linear Markov Decision Processes (where the transition function and reward function can be expressed using a known linear kernel and two unknown vectors). The paper introduces a novel algorithm for finding policies according to a lexicographic objective with bounded regret. While prior work has studied multi-objective optimisation in linear MDPs and lexicographic objectives in finite MDPs, no prior work has considered lexicographic objectives in linear lexicographic MDPs (which generalise finite MDPs). The paper also proves a PAC regret bound (in terms of each objective separately) for their algorithm, which may be the first regret bound for lexicographic MDPs. This bound assumes unknown (& linear) transition dynamics and reward. The paper also bounds regret when an MDP can be approximately expressed as a linear MDP.

## Update after rebuttal
I'm grateful to the authors for including additional information about, e.g., time/space complexity. I think this information significantly improves the paper. Given the complexity of the algorithms presented and the absence of real implementation, I don't feel I can increase my score to a 5, although I still believe the paper should be accepted.

**Claims And Evidence:**

All claims in the submission are made as formal statements and are supported by proofs in the appendix (see below).

**Essential References Not Discussed:**

Although there are a number of other papers exploring lexicographic multi-objective RL (e.g., https://arxiv.org/abs/2408.13493), I don’t know of any other results that attempt to prove regret bounds.

**Experimental Designs Or Analyses:**

There are no empirical experiments in this paper.

**Methods And Evaluation Criteria:**

The only method, Algorithm 1, is justified in terms only of the regret bound. The regret bound is an appropriate criterion if the work is viewed as foundational and is meant to lead to future algorithms that could be applied.
The paper does not state a specific application (a few potential applications of RL in general are mentioned); however, evaluation of the algorithm’s suitability for even toy applications is not given.  For example, there is no analysis of the space or time complexity of the algorithm nor empirical application to a toy lexicographic linear MDP (see weaknesses below).

**Other Comments Or Suggestions:**

I found very few typos in the paper. Example 1 says “arm” (which is not used elsewhere) instead of “action”, but in some contexts, these words are synonymous, so this did not impede clarity.

In the statement of theorems 1 and 2, it is not immediately clear whether the events that each of the m objective regrets satisfy the bound (with probability 1-2\delta) are co-occurring.

**Other Strengths And Weaknesses:**

Strengths:
1. The paper is mostly presented with excellent clarity despite the technicality of its content.
2. The regret bound is, to the best of my knowledge, novel and requires no additional assumptions when compared with prior work (see weakness 1).

Weaknesses:
1. To aid with comparison to existing work investigating lexicographic objectives (not-necessarily-linear) MDPs (such as Skalse et al. 2022), a more explicit discussion of the relationship between MDPs and linear MDPs would be beneficial. For example, it could be clearly stated that finite MDPs can always be represented as linear MDPs with d=|S|\times|A|, as is shown in Example 2.1 of Jin et al. 2020.
2. The space and time complexity of Algorithm 1 is not discussed. I am uncertain, but it is possible that the complexity of this algorithm would make it intractable for real applications. For example, computing lines 20-23 of Algorithm 1 appears to require storing H * K state-action pairs in memory and then computing a least-squares estimate over K values. Although this could be competitive with similar algorithms with regret bounds (in the single-objective case), it may not be scalable to real environments. An explicit readers to understand any (potential) limitations to application.
3. Compounding on the above, the paper does not implement the proposed algorithm and empirically demonstrate how well it performs in comparison to prior literature (which do conduct empirical analysis).

**Questions For Authors:**

No further questions.

**Relation To Broader Scientific Literature:**

The results are well-connected to regret-bounded algorithms in single-objective RL. As far as I know, this is the first result bounding the regret of an algorithm for lexicographic RL.

A number of other papers propose lexicographic RL algorithms without regret bounds. Therefore, a full understanding of the key contributions of this paper may require an empirical comparison.

**Theoretical Claims:**

I reviewed the major results presented in the appendix, and, to the best of my understanding, there are no problems with the correctness of the proofs.

---

> ### Author Rebuttal · Authors · 2025-03-31
>
> We sincerely appreciate your thorough and constructive feedback. We have carefully considered each weaknees and present our responses below, which will be incorporated into the revised paper.
>
> ---
>
> *W1. To aid with comparison to existing work investigating lexicographic objectives (not-necessarily-linear) MDPs (such as Skalse et al. 2022), a more explicit discussion of the relationship between MDPs and linear MDPs would be beneficial.*
>
> Thank you for highlighting this important connection. We have clarified the relationship between general and linear MDPs as follows:
>
> 1. Any finite MDP with state space $\mathcal{S}$ and action space  $\mathcal{A}$ can always be represented as a linear MDP by encoding each state-action pair $(x, a)$ as a one-hot feature vector in $\mathbb{R}^d$, where $d=|\mathcal{S}| \times |\mathcal{A}|$. The transition kernel $\mathbb{P}(x'\mid x, a)$ and reward function $r(x, a)$ can be expressed as inner products between the feature vector of $(x, a)$ and learnable parameters.
> 2. While linear MDPs impose structure that enables tractable theoretical analysis (e.g., regret bounds in Jin et al. 2020), general MDPs with lexicographic objectives (as in Skalse et al. 2022) may not always adhere to this linearity. Our results for linear MDPs directly apply to finite MDPs, but lexicographic objectives in non-linear MDPs may require different algorithms.
>
> ---
>
> *W2. The space and time complexity of Algorithm 1 is not discussed $\cdots$ An explicit readers to understand any (potential) limitations to application.*
>
>
> We thank the reviewer for the constructive feedback. Below, we provide a detailed complexity analysis of Algorithm 1 and discuss its limitations:
>
> 1. Step 7: Computational complexity is $O(d^2|\mathcal{A}|)$, while memory complexity is $O(|\mathcal{S}||\mathcal{A}|)$ for storing state-action pairs.
>
> 2. Step 8: LAE requires $O(md|\mathcal{A}|)$ computations and $O(|\mathcal{S}||\mathcal{A}|)$ memory.
>
> 3. Step 20: Both computational and memory complexity are $O(d^2)$, as $U_h$ can be updated incrementally.
>
> 4. Step 21: Computational complexity is $O(mk)$ for updating $m \cdot k$ values.  Memory complexity is $O(mk+md)$ to store $\\{\hat{r}\_{\tau,h}^i, (x\_{\tau,h+1}, a\_{\tau,h+1})\\}\_{\tau\in[k]}^{i\in[m]}$ and $\\{\hat{w}\_h^i\\}\_{i\in[m]}$.
>
> 5. Step 22: Computational complexity is $O(mkd+md^2)$, dominated by inverting $U_h$ ($O(d^2)$) and computing $m$ linear regressions ($O(mkd+md^2)$). Memory complexity is $O(md)$ for storing weight vectors $\\{\hat{w}\_h^i\\}\_{i\in[m]}$.
>
> 6. Step 23: No additional resources are required, as Q-values can be updated directly from retained $\{\hat{w}_h^i\}_{i\in[m]}$.
>
> Summing across all $H$ MDP layers:
>
> - The computational complexity of is $O(Hmd|\mathcal{A}|+Hd^2|\mathcal{A}|+Hmkd+Hmd^2)$ .
> - The memory is $O(Hd^2+Hmk+Hmd+|\mathcal{S}||\mathcal{A}|)$.
>
> Summing over $K$ rounds:
>
> - The **total computational complexity** is $O(KHd|\mathcal{A}|(m+d)+K^2Hmd+KHmd^2)$ .
> - The memory of our algorithm is $O(Hd^2+HmK+Hmd+|\mathcal{S}||\mathcal{A}|)$.
>
> The $O(K^2)$ computational complexity and $O(K)$ memory are much more expensive than standard bandit algorithms, which typically achieve $O(K)$ computation and $O(1)$ memory. However, our approach remains competitive with existing methods in single-objective MDPs (Jin et al. 2020; Zanette et al. 2020; He et al. 2023). In the revised paper, we will explicitly discuss the computational and memory complexities of our method to clarify its practical applicability.
>
> ---
>
> *W3. Compounding on the above, the paper does not implement the proposed algorithm and empirically demonstrate how well it performs in comparison to prior literature (which do conduct empirical analysis).*
>
> Thank you for raising this issue. The absence of empirical validation in our work aligns with the foundational single-objective linear MDP studies (Jin et al. 2020; Zanette et al. 2020; He et al. 2023), which similarly omit experiments due to challenges in constructing valid linear MDP benchmarks (e.g., enforcing low-rank dynamics and linear payoff structures). We plan to address this limitation in two phases: i) Synthetic experiments will be added to illustrate key theoretical properties. ii) Comprehensive empirical comparisons against heuristic baselines and ablations will be conducted in follow-up work. We appreciate your feedback and welcome suggestions for specific experimental protocols or baseline implementations.
>
> ---
>
> *W4. **Other Comments Or Suggestions:** In the statement of theorems 1 and 2, it is not immediately clear whether the events that each of the m objective regrets satisfy the bound (with probability 1-2\delta) are co-occurring.*
>
> Many thanks for your detailed reviews. The events that each of the $m$ objective regret satisfy the bound are co-occuring. We have polished the paper to avoid typos and improve the clarity in the revised version.

---

### Official Review · Reviewer_a9vt · 2025-03-14

**Overall Recommendation:** 4

**Summary:**

This paper provides an algorithm for the multi-agent reinforcement learning (MORL) setting and regret bounds. Notably the regret bounds given match the single-objective setting up to the leading order term.

## update after rebuttal

I found the work to be making a substantial contribution.  I am glad to see the authors considering the removal of contribution 4 and to tone down the primacy claims.  I think the paper makes a significant contribution without over-claiming or exaggerating.  I think this paper would make a nice contribution to the conference.

**Claims And Evidence:**

The paper explores how techniques in the finite horizon linear MDP setting can be applied to the space of the lexicographically ordered objectives. The paper claims four contributions: A MORL algorithm, a regret bound for the algorithm, a regret bound for the algorithm in the misspecified MORL setting, and a claim of being the first MORL algorithm with a regret bound.

Contributions 1-3 are well supported and insightful.   However, the finite-horizon assumption shouldn't be glossed over as just part of the formalization of the policy 3 pages in, but should be stated much earlier in regards to contextualizing the contributions.

Contribution 4 seems disingenuous. Since the authors’ work is limited to linear MDPs, I don't know what it means to be the "first theoretical regret bound for MORL". In a pedantic sense, it's not true: consider MDPs with a single state and action, all algorithms have zero regret, so there’s a theoretical bound. What about lexicographic ordering in bandits (e.g., Huyuk et al., 2019; Xu & Klabjan, 2023; Xue et al., 2024; all just from a cursory Google search)? Isn't that just a restricted class of MDPs like the authors' restriction? I would strongly recommend dropping this bullet point, and just roll the observation of the sqrt(K) term into one of the other stated contributions (which is the main defensible statement in the contribution).

**Essential References Not Discussed:**

I don't think there's an essential reference missing, but I hope the authors consider being more careful in discussing the historical line of work.

**Experimental Designs Or Analyses:**

NA.

**Methods And Evaluation Criteria:**

Yes.  However this is marred by what seems like overclaiming and exaggeration of the results.

For example, "All of the aforementioned MORL studies primarily rely on empirical evaluations, with limited attention given to theoretical guarantees. This absence of formal analysis has impeded the development of principled algorithms with provable performance bounds." This feels needlessly pejorative. It is true that these past works did not report a regret bound on linear MDPs, but many did some form of formal analysis. Not providing regret bounds does not mean they are not theoretically grounded. For example, Gabor et al. (1998) provided convergence results. How is that not a theoretical guarantee?  The second statement is nonsensical and more pejorative; essentially it’s saying that the lack of algorithms with provable performance bounds has impeded the development of algorithms with provable performance bounds.

Table 1 I find to also be disingenuous. Why is Xu (2020) a row in the table when it introduced an algorithm for finding the Pareto-optimal frontier, which is an entirely different problem? Even Skalse (2022) seems odd as it makes no linear MDP assumptions. These rows seem to be needlessly propping up the authors' work by critiquing others that had different goals.

The paper makes a valuable contribution without this need to oversell their own results and minimize the results of others.  What seems a more accurate description of what's going on is that the authors are using recent techniques for the only-recently explored space of linear MDPs and establishing how they extend into the space of the lexicographic ordering objective.  This is interesting research and deserving to be disseminated without any need to exaggerate the contribution.

**Other Comments Or Suggestions:**

* Line 189: "while still maintaining the feasibility of optimizing": the word choice of "feasibility" seems odd here. Maybe a better wording would be "while still allowing some optimization of".
* Line 301: "is a \epsilon-approximate" shouldn't it be "is an \epsilon-approximate"?
* Line 371: "Next, the agent proceeds to eliminate arms based on the second objective." First time the word "arms" is used in the paper. I understand, but this is adding confusion.
* Line 373: "which is disappointing because a_3 is awful for the third objective." Isn't it disappointing because a_3 is not the optimal action (since 4 < 5)?
* The term "the lexicographically optimal policy..." is used in several places.  The policy is likely not unique, so maybe this should say "A lexicographically optimal policy..."
* Why is lexicographic ordering defined as "dominates"? Dominance I would expect to be reserved for a partial order, not the total order of lexicographic ordering.
* I believe there are typos in the definitions of the action value functions. Throughout the paper, x is used to denote state however in the value function definitions, reward is written as a function of s.
* There is also a typo in Assumption 1 that is critical to fix. I believe the superscript of the reward terms should be a j. As stated, the inequality does not make sense.
* Brackets around the transition-kernel value-function product before/in Equation (1) is confusing given the meaning of brackets as a set.

**Other Strengths And Weaknesses:**

I really like section 6. I find it helpful to understand what's going on. However, I feel like Section 6 is out of order. Shouldn't this come before Section 4, as it motivates the underlying techniques employed in the algorithm? Or at least 6.1 should come first?

Some additional discussion of where  assumption 1 comes from would be nice. This assumption seems super restrictive! Might it actually allow the objective to satisfy the Continuity axiom of von Neumann and Morganstern? This would then admit a single scalar reward signal that would allow maximizing its expectation reducing the entire problem to traditional RL. For finite states and actions, there always exists such a lambda, and so this assumption just creates a constant to use in the bound. However, what if the action space allows for continuous probabilities such as admitting stochastic policies, this seems to rule out certain discontinuities that would naturally arise with lexicographic orderings and are what makes them challenging to begin with (e.g., see Gabor et al., 1998; Bowling et al., 2023).

I think the introduction of linear RL in Section 2 should include some discussion of what "linearity in both the reward function and transition probabilities" means, and a definition of “misspecified”. The current related work section has no real value. It's merely a list of recent papers and the resulting regret bound. The form of the problem addressed (e.g., exactly what is linear and how) would seem way more important than the regret bound form itself. After all, your main deviation is in the form of the problem to explore a lexicographic objective.

**Questions For Authors:**

None.

**Relation To Broader Scientific Literature:**

This is one of the weaknesses of the paper.

According to the introduction, MORL started in 2013, but no interesting advance happened until this decade.

Yet the cited 2013 paper is, in fact, a survey with over a hundred citations of work in this area going back decades. Lexicographic ordering in RL goes back at least to Gabor et al. (1998), which (to be fair) is discussed eventually on page 3, but long after the introduction seems to disregard this history.

I would say this is doing a poor job of placing the contributions in the broader scientific literature.

**Theoretical Claims:**

I mostly followed the definition of the algorithm and the approach all seems plausible, but I did not verify any of the proofs (all in the supplementary material).

---

> ### Author Rebuttal · Authors · 2025-03-31
>
> We sincerely appreciate the constructive feedback and have carefully considered the raised concerns. Our point-by-point responses follow below.
>
> ---
>
> *Q1. **Claims And Evidence:** Contributions 1-3 are well supported and insightful. However, the finite-horizon assumption shouldn't be glossed over $\cdots$.*
>
> We appreciate the reviewer’s positive feedback on Contributions 1-3 and their helpful suggestion. We agree that introducing the finite-horizon assumption earlier in the paper will better clarify our work’s scope. In the revised version, we will highlight this in the Introduction.
>
> ---
>
> *Q2. **Claims And Evidence:** Contribution 4 seems disingenuous $\cdots$ Isn't that just a restricted class of MDPs like the authors' restriction?*
>
>  We agree that calling our work the "first theoretical regret bound for MORL" was inaccurate, especially since prior work on lexicographic bandits  (e.g., Huyuk et al., 2019), which is a special case of MDPs with $H=1$.  We will remove Contribution 4 and instead integrate the discussion of the $\sqrt{K}$ regret term into our other contributions. Meanwhile, we will add comparisons to bandit-based MORL frameworks (Huyuk et al., 2019) in the related work section.
>
> ---
>
> *Q3. **Methods And Evaluation Criteria:** Why is Xu (2020) a row in the table $\cdots$ Even Skalse (2022) seems odd as it makes no linear MDP assumptions.*
>
> To the best of our knowledge, no prior work specifically tackles multi-objective linear MDPs, so we refer to general multi-objective MDP (MOMDP) frameworks. Xu et al. (2020) focuses on finding Pareto-optimal frontiers in MOMDPs under Pareto ordering. Skalse et al. (2022) studies lexicographic ordering in MOMDPs without assuming linearity. We will update the table by removing Xu et al. (2020) and Skalse et al. (2022), retaining only works directly relevant to **linear MDPs** to avoid confusion.
>
> ---
>
> *Q4. **Relation To Broader Scientific Literature:**  Lexicographic ordering in RL goes back at least to Gabor et al. (1998), which (to be fair) is discussed eventually on page 3, but long after the introduction seems to disregard this history.*
>
> We thank the reviewer for their helpful comment on the history of lexicographic ordering in RL. We will restructure the introduction to foreground the seminal work of Gabor et al. (1998) as the conceptual origin of lexicographic RL so as to strengthen the paper’s scholarly context.
>
> ---
>
> *Q5. **Other Strengths And Weaknesses:** Some additional discussion of where assumption 1 comes from would be nice.*
>
> Assumption 1 addresses the Optimal Action Preservation Dilemma (**Section 6**). In Example 1, there are three Q-value vectors: $[5,5,5], [1,5,5]$ and $[4,10,1]$ for actions $a_1, a_2$ and $a_3$, where $\lambda=\frac{10-5}{5-1}=5$. $a_1$ is lexicographically optimal. When eliminating actions based on the first objective, $a_2$ is eliminated since $1$ is far from $5$, but $a_3$ is kept as $4$ is close to $5$, leaving $\\{a_1,a_3\\}$. Next, elimination considers the second objective. Although $10$ (from $a_3$) is much bigger than $5$ (from $a_1$), the confidence term $\beta_k\cdot C$ is scaled by $2+4\lambda$ (Step 4 of Algorithm 2), ensuring $a_1$ stays in $A_s^2$.
>
> ---
>
> *Q6. **Other Strengths And Weaknesses:** Might it actually allow the objective to satisfy the Continuity axiom of von Neumann and Morganstern?*
>
> The continuity axiom says that for three options where $A \succ B \succ C$ , some mix of $A$ and $C$ should be equally good as $B$. This means no outcome is infinitely better or worse than another. But in lexicographic optimization, higher-priority goals (like safety) are infinitely more important than lower ones (like cost). Thus, the continuity axiom may not apply, since no trade-off can make such different-priority goals equivalent.
>
> ---
>
> *Q7. **Other Strengths And Weaknesses:** I think the introduction of linear RL in Section 2 should include some discussion of what "linearity in both the reward function and transition probabilities" means, and a definition of “misspecified”.*
>
> We will revise Section 2 to clarify the concepts of "linearity in both the reward function and transition probabilities" and discuss "misspecified." Specifically, we will explain that in linear RL the reward function is $r_h(x,a) = \phi(x,a)^\top \theta_h$, where $\phi(s,a)$ is a known feature vector and $\theta_h$ is unknown. The transition probabilities follow $\mathbb{P}_h(x'|x,a)=\phi(x,a)^\top \mu_h(x')$, where $\mu_h(x')$ is an unknown measure. Additionally, we will clarify that "misspecified" refers to settings where the true environment deviates from the assumed linear class (e.g., due to approximation errors in rewards or transitions).
>
> ---
>
> *Q8. **Other Comments Or Suggestions.***
>
> We sincerely appreciate the reviewer’s detailed feedback, which has significantly contributed to improving the quality of our paper. All suggested revisions have been carefully considered and will be incorporated into the final version of the paper.

---

### Decision · Program_Chairs · 2025-05-01

**Decision:**

Accept (poster)

**Comment:**

This paper presents a novel algorithm for multi-objective reinforcement learning (MORL) under lexicographically ordered objectives in the linear MDP setting. The proposed method, Lexicographic Linear Reinforcement Learning (LLRL), provides regret guarantees that match the single-objective case up to leading-order terms and extend to the misspecified setting. The work offers a rigorous analysis of a challenging objective structure where Bellman optimality does not naturally hold and introduces a principled action elimination mechanism to handle the lexicographic constraints.

The paper's main strengths include its theoretical contributions, well-structured algorithmic design, and clear motivation for exploring lexicographic preferences within the linear MDP framework. The reviewers praised the novelty of formal regret bounds in this space and the careful use of linear assumptions to enable tractable analysis. However, several concerns were raised. One reviewer criticized the initial claim of being the first MORL regret bound as overstated, particularly in light of prior work in lexicographic bandits. Another reviewer highlighted the strength of Assumption 1, questioning whether it might reduce the problem to scalarized RL. In addition, the absence of experimental results and the lack of discussion around computational complexity were noted as limitations. Some confusion also stemmed from the paper’s notation and incomplete treatment of earlier MORL literature.

The authors responded constructively to all these points. They agreed to remove and rephrase the overstatement of contribution, clarified their position concerning prior bandit literature, and significantly improved the theoretical basis by revising Assumption 1. This revised assumption, introduced in response to a key concern about Lemma 3, was positively acknowledged by the reviewer who raised it. A subsequent technical exchange regarding scalarization-based strategies led to a meaningful theoretical comparison, in which the authors convincingly demonstrated the advantages of their approach in terms of regret bounds for high-priority objectives. Further, the authors provided a complete computational complexity analysis and outlined a clear plan for future empirical validation, consistent with precedent in theoretical RL papers.

Overall, the submission makes a timely and rigorous contribution to MORL. In addition, the authors’ technically deep responses helped resolve key concerns.